# Ventral pallidum encodes relative reward value earlier and more robustly than nucleus accumbens

David Ottenheimer [1], Jocelyn M. Richard [2,4] & Patricia H. Janak [1,2,3]

The ventral striatopallidal system, a basal ganglia network thought to convert limbic information into behavioral action, includes the nucleus accumbens (NAc) and the ventral pallidum (VP), typically described as a major output of NAc. Here, to investigate how reward-related information is transformed across this circuit, we measure the activity of neurons in NAc and VP when rats receive two highly palatable but differentially preferred rewards, allowing us to track the reward-specific information contained within the neural activity of each region. In VP, we find a prominent preference-related signal that flexibly reports the relative value of reward outcomes across multiple conditions. This reward-specific firing in VP is present in a greater proportion of the population and arises sooner following reward delivery than in NAc. Our findings establish VP as a preeminent value signaler and challenge the existing model of information flow in the ventral basal ganglia.

[1] Solomon H. Snyder Department of Neuroscience, Johns Hopkins School of Medicine, Johns Hopkins University, 725 N. Wolfe St., Baltimore, MD 21205, USA. [2] Department of Psychological and Brain Sciences, Krieger School of Arts and Sciences, Johns Hopkins University, 3400 N. Charles St., Baltimore, MD 21218, USA. [3] Kavli Neuroscience Discovery Institute, Johns Hopkins University, 3400 N. Charles St., Baltimore, MD 21218, USA. [4] Present address: Department of Neuroscience, University of Minnesota, 321 Church St. SE, Minneapolis, MN 55455, USA. Correspondence and requests for materials should be addressed to D.O. (email: david.ottenheimer@jhu.edu) or to P.H.J. (email: patricia.janak@jhu.edu)

A daptive reward consumption requires proper valuation of each rewarded outcome relative to all available options in order to select the appropriate consummatory response. One circuit frequently implicated in such reward-related processing is the ventral striatopallidal system. This basal ganglia network, primarily composed of the nucleus accumbens (NAc) and ventral pallidum (VP), receives dense inputs from limbic structures, such as hippocampus, prefrontal cortex, and amygdala[1–3], leading to the hypothesis that this system is a critical interface between reward processing and motor output[4–7]. Accordingly, many reward-related behavioral responses depend on normal connectivity between NAc and VP[8–11]. Together, these results and the anatomical positioning of NAc and VP within the basal ganglia have contributed to the notion that, within this circuit, the main purpose of VP is to pass on reward-related limbic and motor information from its striatal partner, NAc[12,13].

A serious limitation thus far in the characterization of the ventral striatopallidal system is the lack of comparative observations in NAc and VP on a timescale relevant for the reward-related processing in which the circuit is functionally implicated. Understanding the transformation of reward-related information across the ventral striatopallidal system during a seconds-long behavioral response requires temporally precise measurements of neural activity in each region. Previous work has identified neurons in both NAc[14–24] and VP[25–32] with phasic responses to rewards (and their predictive stimuli) that track outcome value, but it is unclear how reward-related neural responses in VP result from activity in NAc, as the classic model of ventral striatopallidal function would predict. In fact, a recent comparison of activity in NAc and VP in the same behavioral task found that the onset of cue responses in VP frequently precedes their onset in NAc[28], leaving the question of whether VP acts exclusively downstream of NAc in this reward processing circuit.

To further interrogate the respective roles of VP and NAc in reward processing, we measured neural activity in a task with multiple reward outcomes. In addition to permitting a comparison of the onset of phasic activity in response to reward outcome, this approach allowed us to track over time the reward-specific information contained within the spiking activity of individual neurons and neural ensembles in each region. Surprisingly, our data indicate that a much greater proportion of VP neurons are reward-selective than neurons in NAc. Moreover, the reward-specific information signaled by both individual neurons and ensembles in VP precedes that signaled by NAc neurons. Further, VP neurons reliably and rapidly track relative value across a variety of reward conditions. The flexibility of this VP value signal and its abundance within the neural population establish VP as a robust value signaler and suggest it does so at least partly independently of its classical input, NAc, encouraging consideration of VP as an important reward processing center rather than simply a relay for reward-related information to motor outputs.

## Results

**Rats prefer sucrose over maltodextrin in home cage and task.** To test encoding of multiple rewards in NAc and VP, we chose to compare responses to 10% solutions of sucrose and maltodextrin, two palatable carbohydrates with equivalent caloric value but distinct tastes[33–35]. After multiple days of free access to the solutions in their home cages, rats began training on the behavioral task. On each trial, 110 µL of reward solution was delivered into a metal bowl contingent upon rats' entry into the reward port during a 10 s white noise cue (Fig. 1a). Trials with presentation of a given solution were pseudorandomly interspersed throughout the session, an approach that obscured the reward identity from the rat until the solution was delivered. Once rats responded to

the cue on 80% of trials, we implanted drivable tungsten electrode arrays in either NAc or VP (Fig. 1b). To evaluate reward preference, we conducted 60-min two-bottle choice tests prior to and following the first and last recording sessions with sucrose and maltodextrin; rats consistently showed a preference for sucrose (Fig. 1c).

Neural recording sessions began after rats recovered from surgery. To monitor consumption during the task, we recorded each rat's individual licks during each recording session. The overall licking pattern was similar for both rewards; there was no significant main effect of reward on the total number of licks (F $(1,3142) = 1.24$, $p = 0.29$) or the total duration of licking (F $(1,3142) = 0.303$, $p = 0.59$) within the 15 s following reward delivery. However, rats licked slightly, but significantly, more for the preferred reward, sucrose, during the period 1–4.5 s following reward delivery (23.2 vs. 22.3 licks; $F(1,3142) = 66.0$, $p = 5.3E-6$; Fig. 1d, Supplementary Fig. 2). Complementarily, the interlick intervals following the first 30 licks of each trial were significantly shorter on sucrose trials (Fig. 1e; $F(1,3000) = 33.3$, $p = 0.000084$). This accelerated consumption of sucrose echoes the rats' preference for sucrose over maltodextrin in the two-bottle choice test (Fig. 1c).

**More neurons in VP fire reward-selectively than in NAc.** We collected neural activity from six rats with electrodes in NAc (182 neurons, 4–49 per rat, median 32, 36 sessions) and five rats with electrodes in VP (436 neurons, 32–137 per rat, median 86, 25 sessions). Neurons in both regions responded to reward-related events: cue onset, port entry (PE), and reward delivery (RD) (Supplementary Fig. 3), consistent with prior findings (NAc:[15–21,24,36,37], VP:[25–28,31,38,39]). To evaluate reward selectivity, we focused on the neural activity following reward delivery, when the rats first detected the identity of the reward and consumed it. Initial inspection of the average firing rates of all neurons during this epoch divided into sucrose and maltodextrin trials revealed greater firing for sucrose among reward-excited neurons in both regions (Supplementary Fig. 4, see also Fig. 5). Evidence of reward-specific firing was also evident in peri-event histograms of individual neurons' spiking in each region (Supplementary Fig. 5).

To more precisely determine the presence and onset of reward-selective responding, we divided the time surrounding reward delivery into overlapping bins with a sliding window of 600 ms advanced by 100 ms. For each bin, we found the number of neurons whose firing rates were significantly differentially modulated across sucrose and maltodextrin trials and categorized these neurons by which reward elicited greater firing. We conducted this analysis for all neurons from each region (Fig. 2a, b), as well as for each individual rat to ensure general consistency across subjects and recording locations (Supplementary Fig. 6). Notably, the peak number of reward-selective neurons in any given bin was greater in VP than in NAc (33% vs. 10%, $\chi^2 = 34.3$, $p = 4.7E-9$) and this bin with peak reward selectivity was earlier in VP (centered at 1.1 s) than in NAc (centered at 1.9 s) (Fig. 2a, b). We compared the time course of selectivity in each region by subtracting the proportion of selective neurons in VP from the proportion in NAc in each bin (Fig. 2c), revealing that at no point was there more reward selectivity in NAc than in VP. We also compared the onset of reward-selective responses in each region and found that the distribution of onsets was earlier in VP than in NAc (Fig. 2d).

To characterize the activity of reward-selective neurons in each region, we identified neurons that met our criteria for reward selectivity in any of the bins centered 0.4–3 s after reward delivery, a period of time that captured the majority of phasic

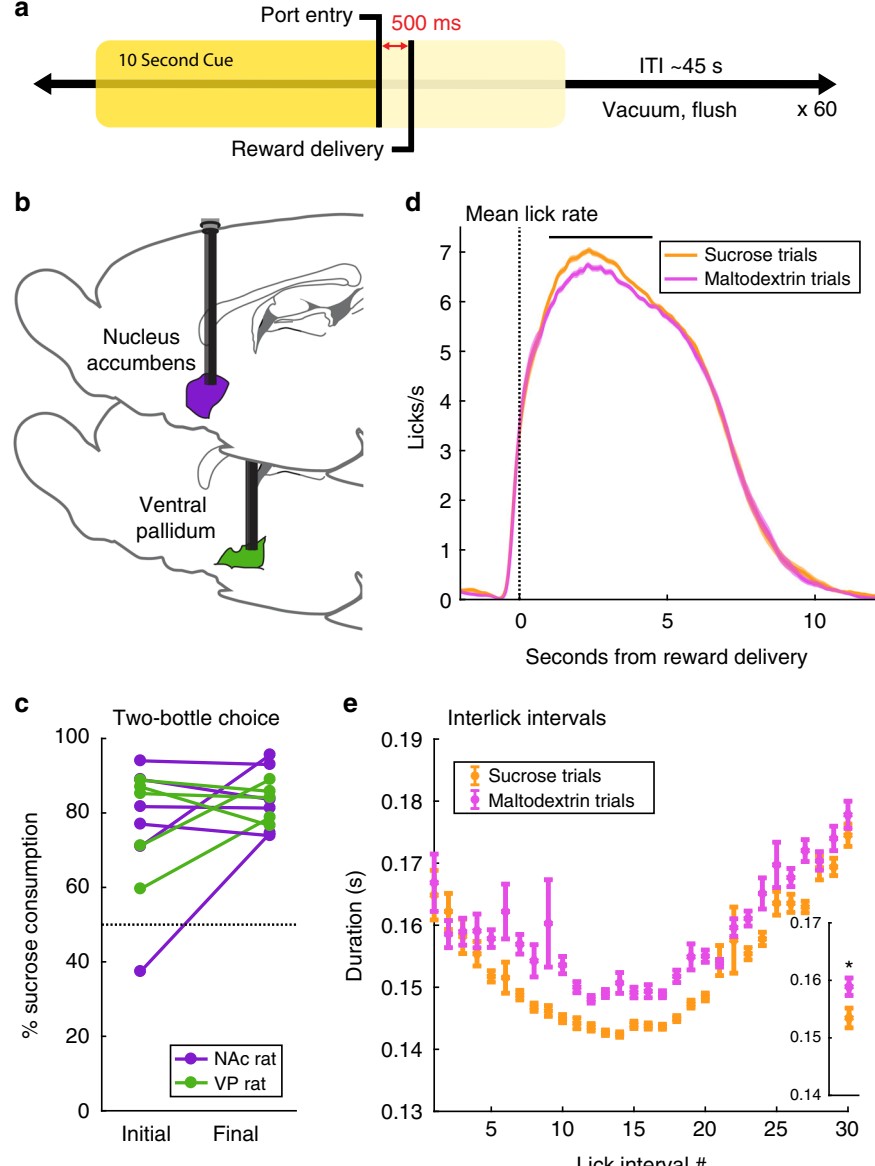

**Fig. 1** Preference for sucrose over maltodextrin in home cage drinking and following cued reward delivery. **a** Task design. Sucrose or maltodextrin solution was delivered 500 ms following rats' entry into the reward port during a 10 s white noise cue. Trials of each reward were randomly interspersed throughout the session such that reward identity was unpredictable to the rat. **b** After training, drivable 16 electrode arrays were implanted in either nucleus accumbens ($n = 6$) or ventral pallidum ($n = 5$). See Supplementary Fig. 1 for placements. **c** Rats' preference (percentage sucrose consumption of total consumption) during 1 h free access to 10% solutions of sucrose and maltodextrin. Tests were after surgical recovery (Initial) and after final session with sucrose and maltodextrin (Final). **d** Average lick rate on sucrose (orange) and maltodextrin (pink) trials during the task. Shading is SEM. Black bar indicates greater number of licks on sucrose trials 1–4.5 s post reward delivery ($F_{(1,3142)} = 66.0$, $p = 5.3\text{E-}6$). See also Supplementary Fig. 2. **e** Interlick interval duration following the first 30 licks on sucrose (orange) and maltodextrin (pink) trials. Inset: mean interlick interval duration across all 30 intervals. Asterisk indicates significant main effect of reward on duration ($F_{(1,3000)} = 33.3$, $p = 0.000084$)

reward-selective responses across both regions (Fig. 2a, b). We then plotted these neurons' individual and averaged activity on sucrose and maltodextrin trials (Fig. 2e–p). Within this time window, 24% of neurons in NAc and 52% of neurons in VP were at one point reward-selective, a significantly greater proportion in VP ($\chi^2 = 39.9$, $p = 2.6\text{E-}10$). In both regions, we found that most of the reward-selective responses were excitations for sucrose; some of these cells were also inhibited for maltodextrin (Fig. 2e–g, k–m). A smaller subset of reward-selective cells in each region had greater firing rates for maltodextrin, often due to an inhibition for sucrose (Fig. 2h–j, n–p). Thus, despite only minimal differences in licking behavior for each reward, a substantial proportion of neurons in both VP and NAc fire in a reward-selective manner, and these reward-selective responses are represented in a greater proportion of the recorded population in VP than in NAc.

**VP neurons decode trial type earlier and more accurately than NAc.** The analysis above indicates differential encoding of two rewards, sucrose and maltodextrin, with most selective units showing greater responding for sucrose, the preferred reward, over maltodextrin. To complement this analysis, we used linear discriminant analysis (LDA) to test when and to what extent

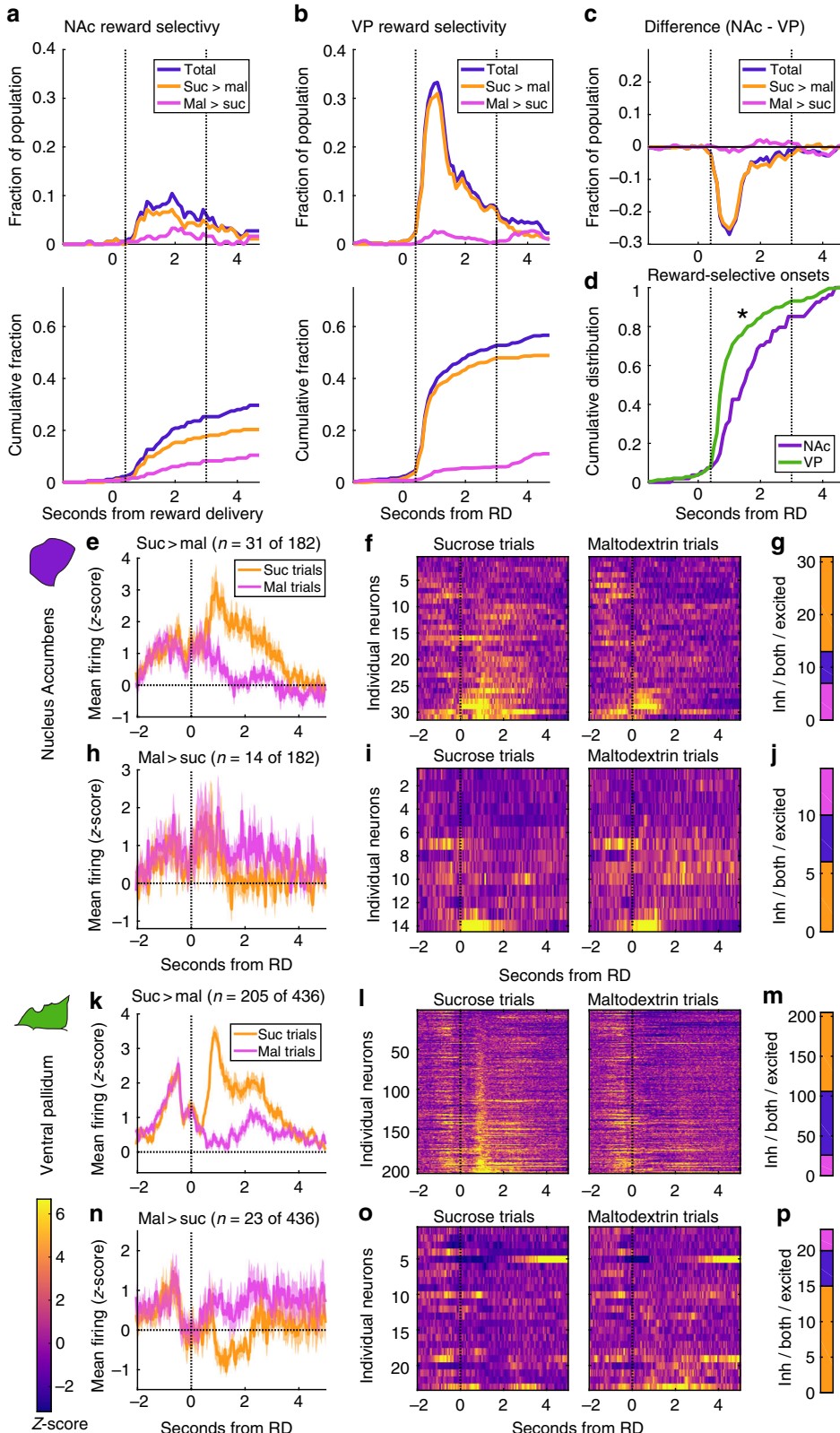

neural activity in each region could be used to predict reward identity. Using fivefold cross-validation, we determined for each 600 ms bin how accurately LDA models trained on the spike activity of individual neurons could classify trials as sucrose or maltodextrin. Models trained on single unit activity classified trial type at rates above chance in both VP and NAc (Fig. 3a).

Focusing on our window of interest from Fig. 2 (0.4–3s post reward delivery), we found that VP single unit accuracy improved over shuffled data more than NAc (shuffled vs true X region: $F_{(1, 31150)} = 11.5$, $p = 0.0019$). When comparing the most accurate bin in NAc (centered at 1.4 s) to that in VP (centered at 1 s), there was a noticeable shift in classification accuracy in the cumulative

**Fig. 2** More neurons in VP fire selectively for sucrose and maltodextrin than in NAc. **a, b** Top panel: fraction of NAc (**a**) and VP (**b**) neurons meeting criteria for reward selectivity as a function of time after reward delivery. Plotted are total fraction of reward-selective neurons (blue) and, of those, neurons with greater firing for sucrose (orange) and greater firing for maltodextrin (pink). Bottom panel: Cumulative distribution of reward selectivity over time after reward delivery. **c** Subtraction of VP reward selectivity from NAc in each bin. Negative values indicate more selectivity in VP. **d** Cumulative distribution of reward selectivity onsets as a fraction of total reward-selective neurons. Asterisk indicates significantly earlier onsets in VP (F(1,290) = 12.7, p = 0.00071). **e–g** Neurons with greater firing for sucrose in any bin centered at 0.4–3 s. **e** Mean normalized firing rate for sucrose-selective neurons on sucrose (orange) and maltodextrin (pink) trials. Shading is SEM. **f** Heat maps of the normalized activity of individual sucrose-selective neurons on sucrose and maltodextrin trials. **g** Number of neurons with maltodextrin inhibitions (pink), sucrose excitations (orange), or both (blue). **h–j** Neurons in NAc with greater firing rate for maltodextrin in any of the bins centered 0.4–3 s. **h** Mean normalized firing rate for maltodextrin-selective neurons on sucrose (orange) and maltodextrin (pink) trials. Shading is SEM. **i** Heat maps of the normalized activity of individual maltodextrin-selective neurons in NAc on sucrose and maltodextrin trials. **j** Number of neurons in NAc with maltodextrin inhibitions (pink), sucrose excitations (orange), or both (blue). **k–p** Sucrose- (**k–m**) and maltodextrin- (**n–p**) selective neurons in VP, plotted as for NAc neurons in **e–j**

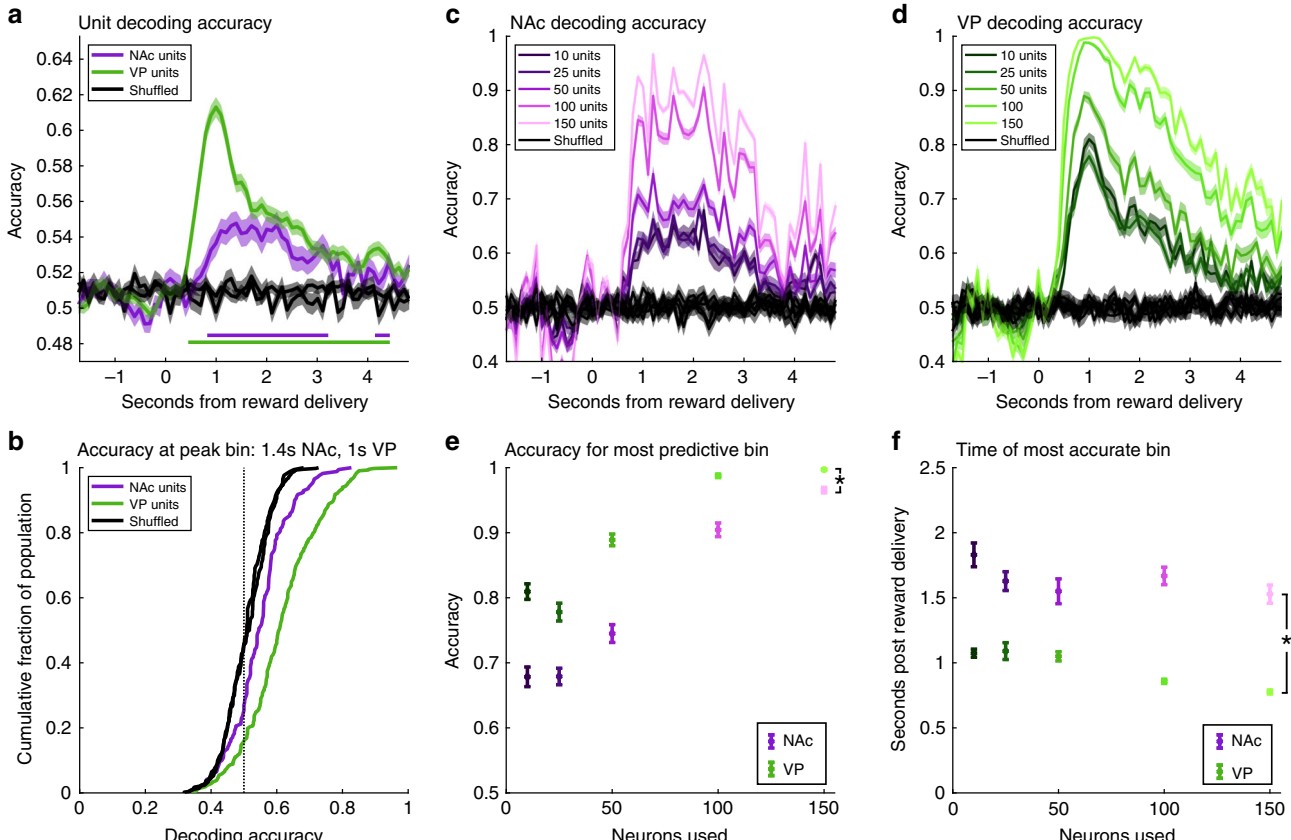

**Fig. 3** VP activity decodes trial identity earlier and more accurately than NAc activity. **a** Average cross-validated decoding accuracy relative to reward delivery time, determined using linear discriminant analysis models trained on spiking data of individual neurons across 600 ms overlapping bins. Decoding accuracy for NAc (purple), VP (green), and data with shuffled trial identity from each region (black). Shading is SEM. Purple (NAc) and green (VP) lines indicate consecutive bins where accuracy exceeds 99% confidence interval of corresponding shuffled data. **b** Cumulative distribution of accuracies in the bin with the greatest average accuracy in each region (centered at 1.4 s in NAc and 1 s in VP) and the corresponding shuffled data from that bin in each region. **c** Average cross-validated decoding accuracy of linear discriminant analysis models trained on spiking data of 20 randomly selected groups of 10, 25, 50, 100, or 150 neurons in NAc across 600 ms overlapping bins relative to reward delivery time and corresponding models trained on data with trial identity shuffled. Shading is SEM. **d** Same as (**c**) for VP pseudoensemble models. **e** Average accuracy of each replicate for the bin with peak accuracy for each pseudoensemble size in each region. Asterisk indicates significant main effect of region on accuracy (F(1,490) = 212, p = 3.3E-40). **f** Average peak accuracy time post-reward for each replicate of each pseudoensemble size in each region. Asterisk indicates significant main effect of region on peak accuracy time (F(1,490) = 289, p = 2.5E-51)

distribution function (CDF) (Fig. 3b), corresponding to a significantly greater improvement in accuracy over shuffled data in VP (shuffled vs true X region: F(1,1154) = 13.6, p = 0.00037). Notably, VP single units first improved over shuffled data for the bin centered at 0.5 s, whereas NAc single units first improved over shuffled data at 0.9 s (purple and green lines in Fig. 3a). To control for the larger number of neurons recorded in VP, we

conducted the analysis 20 more times with 182 randomly chosen VP units. The first bin significantly more accurate than shuffled data ranged from 0.4–0.6 s (median 0.5 s), consistently earlier than 0.9 s in NAc.

Although the data from individual neurons points to more reward-selective activity in VP than NAc, an alternate explanation is that reward-specific information is more distributed across

neurons in NAc than in VP. If so, including additional neurons in the model should improve the accuracy of the NAc decoders relative to VP. To overcome the limited number of sessions in NAc with greater than five neurons, we pooled neurons together into pseudoensembles to compare how much information is contained within larger groups of neurons in each region. We ran the same analysis as before using LDA models trained with the spiking activity of 10, 25, 50, 100, and 150 neurons randomly selected from each region. Increasing the number of neurons improved accuracy in both regions (Fig. 3c, d), contributing to a significant main effect of ensemble size on peak bin accuracy (F (4,490) = 237, p = 4.3E-113). Pseudoensembles in VP had greater peak accuracy than those in NAc across all levels, evident in a main effect of region on peak bin accuracy (F(1,490) = 212, p = 3.3E-40; Fig. 3e). Notably, pseudoensembles in VP reliably reached 100% decoding accuracy with 100 neurons; NAc pseudoensembles reached at most 97% with 150 neurons (Fig. 3e). The smaller difference in accuracy between the two regions with 150 neurons was reflected in a significant interaction between ensemble size and region on decoder accuracy (F(4,490) = 8.73, p = 8.2E-7). Even at larger sizes, VP pseudoensembles consistently achieved peak accuracy earlier than those in NAc (Fig. 3f; main effect of region: F(1,490) = 289, p = 2.5E-51). Overall, our results from these decoding analyses confirm that VP neurons contain more reward-specific information than NAc neurons and indicate that this information arises and peaks earlier in VP than in NAc.

While our initial decoding analysis included all neurons from each region regardless of their status as reward-selective or not, we were also interested in directly comparing the amount of reward-specific information contained in the reward-selective population in each region, so we conducted the same decoding analyses but restricted our sample to those neurons classified as reward-selective in Fig. 2. The accuracy of single unit models trained exclusively on reward-selective neurons was much closer across regions (Supplementary Fig. 7a); VP no longer improved over shuffled data more than NAc in the window 0.4–3 s post reward delivery (shuffled vs. true X region: F(1, 13612) = 0.0952, p = 0.7617) nor when comparing the peak bin in each region (shuffled vs true X region: F(1, 504) = 3.46, p = 0.080). Nevertheless, there continued to be a noticeably earlier rise and peak in accuracy for VP models than for NAc (Supplementary Fig. 7a). For pseudoensemble models, we were limited by the number of reward-selective neurons in NAc, but we found that with groups of 10 and 25 neurons, there was a main effect of region on peak bin accuracy (F(1,196) = 38.9, p = 2.7E-9; Supplementary Fig. 7e) and time of peak accuracy (F(1,196) = 54.8, p = 3.8E-12; Supplementary Fig. 7f), indicating that VP pseudoensembles consisting of reward-selective neurons are more predictive and achieve peak accuracy earlier than NAc reward-selective pseudoensembles. Overall, these data provide evidence that, even among the reward-selective population, VP neurons represent reward-specific information earlier and more strongly than NAc.

**VP reward signal reflects previous outcome**. Because the predominant reward-selective response in both regions was increased spiking for the preferred reward (sucrose) relative to maltodextrin, and given the results from previous recording studies[15–17,19,21,22,25,26,31,32,40], we hypothesized that this reward-specific signal reflects relative reward value. If so, we would predict that the report of relative value would depend on recent reward history, which we can approximate by analyzing trials according to both the current and previously received reward. For instance, the relative value of sucrose would be greater following trials where rats received maltodextrin, and maltodextrin's value would be lesser following sucrose trials. To look for evidence of

such a scheme, we plotted the mean activity of all neurons in NAc and VP for each of the four combinations of previous and current reward (Fig. 4). While there was some evidence for previous-reward modulation of reward response across the population of neurons in NAc (Fig. 4a, b), VP neurons showed very prominent modulation of the reward-related response according to our prediction: greater firing for sucrose following maltodextrin trials and lesser firing to maltodextrin following sucrose (Fig. 4c, d). When analyzing the contribution of reward and previous reward to the neural activity in each region 0.8–1.3 s following reward delivery, we found a significant main effect for previous reward in VP (F(1,1724) = 10.1, p = 0.022) but not in NAc (F(1,704) = 0.0167, p = 0.90), though a test including data from both regions did not find a significant interaction between previous reward and region (F(1,2428) = 3.89, p = 0.055).

We next sought to more quantitatively assess the impact of previous outcomes on the reward-evoked signals in each region by using a linear model approach that predicts a neuron's firing rate based on the reward outcomes on the current trial and each of the prior six completed trials[41]. The weights of the coefficients assigned to each trial reveal how strongly the outcome from that trial factors into the neuron's firing rate on the current trial. For both the current trial and the previous trial, only VP models showed, on average, coefficients that deviated from chance (Fig. 4e). Consistent with our relative value hypothesis and with our observations in Fig. 4c, d, the direction of the coefficients indicated a strong positive impact of receiving sucrose on the firing rate in the current trial and a negative impact of sucrose received on the previous trial. We also found that more neurons in VP had significant coefficients than in NAc for both the current and most recent trial (Fig. 4f). We found no impact of previous trials beyond the most recent on reward-related firing in either region. We also conducted this analysis on cue- and port entry-evoked firing in NAc and VP; surprisingly, we found that receiving sucrose on either of the previous two trials had a positive impact on the port entry-related firing in VP, and this effect was greater in reward-selective neurons (F(1,852) = 10.1, p = 0.024; Supplementary Fig. 8). Thus, firing in VP at both the time of the reward and the reward-seeking action reflects the recent reward history.

**VP signals reward value relative to currently available options**. Our results comparing VP neural responses to sucrose and maltodextrin are consistent with a relative value signal, but a stronger test of this hypothesis requires changing the relative value of the reward outcomes and looking for a corresponding change in neural activity. Such an approach has demonstrated that neurons in NAc report relative value[21,22,42], but it is unclear whether VP neural reports of a reward's value are relative to other currently available outcomes. We tested for relative value by conducting an additional session for VP rats in which sucrose was replaced with water, an outcome much less rewarding than both sucrose and maltodextrin solutions[34,43]. We predicted that, if VP neural activity reflects relative value, then the predominant reward-specific neural response would be excitations for maltodextrin, which in this scenario is the preferred outcome. Alternatively, if VP activity reflects absolute value, then the neural responses would remain suppressed to maltodextrin as in the sessions with sucrose and maltodextrin (Fig. 2k).

Two rats successfully completed this session type, contributing a total of 125 neurons (79 and 46, respectively). Water was much less preferred than maltodextrin, evident in the mean lick rate for each outcome (Fig. 5a). By calculating the number of reward-selective neurons across 600 ms bins, we saw an even greater proportion of neurons in VP showed reward-specific responses

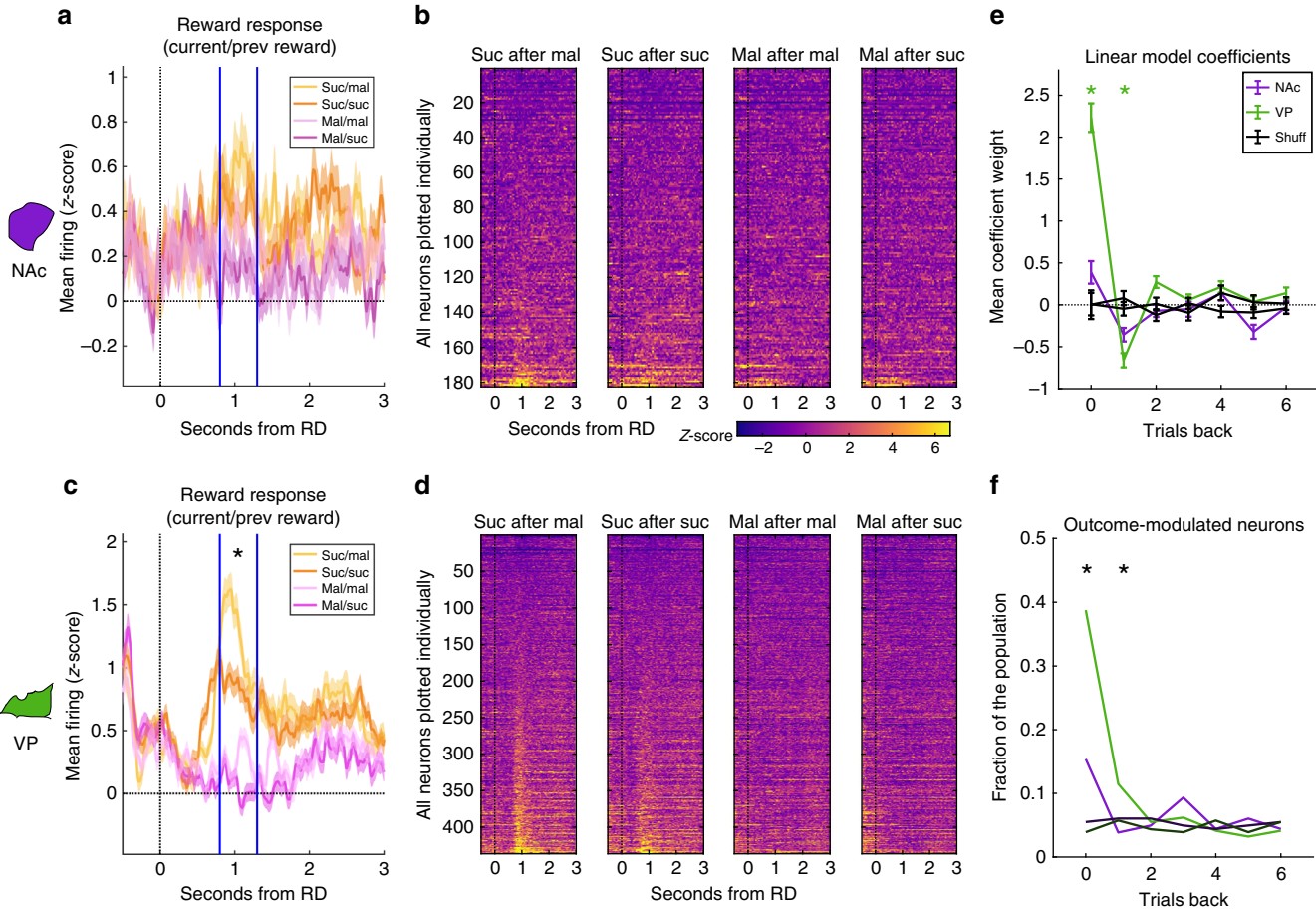

**Fig. 4** Previous reward outcome impacts current reward firing. **a**, **c** Normalized activity of all neurons in NAc (**a**) and VP (**c**) on sucrose (orange) and maltodextrin (pink) trials of each reward, separated by reward outcome on preceding trial; darker lines indicate that sucrose was the prior trial's reward. Asterisk indicates a significant main effect of previous reward on normalized firing rate in VP ($F(1,1724) = 10.1$, $p = 0.022$) for the epoch bounded by the vertical blue lines. **b**, **d** Normalized reward-related activity of every individual neuron in NAc (**b**) and VP (**d**) on trials with each combination of previous and current reward. **e** Mean coefficient weights for the impact of the current and previous 6 trials on normalized firing rate in the same epoch as (**a**, **c**) for each neuron in NAc (purple), VP (green), and corresponding data for each neuron with the outcomes shuffled (black) for each region. Error bars are SEM. Asterisks are $p < 0.05$ for Tukey tests comparing VP coefficients to shuffled data, corrected for multiple comparisons. **f** Proportion of the neural populations in VP (green), NAc (purple), and corresponding shuffled neurons (black) with significant coefficients for each of the relative trials. Asterisks represent $p < 0.05$ for chi-square tests on both the distribution of neurons across all four conditions (true and shuffled data from each region) and across the true data from each region

for any given bin (59% at 1.3 s) (Fig. 5b). We plotted the activity of neurons that met our criteria for reward selectivity during any bin within the 0.4–3 s window we used in Fig. 2 (Fig. 5c, d); 70% of neurons were reward-selective during this time, a significantly greater proportion than the 53% in sessions these two rats completed with sucrose and maltodextrin ($\chi^2 = 9.68$, $p = 0.0019$). This higher proportion may reflect the considerable difference in value between water and maltodextrin compared to the similar value of the two appetitive reward outcomes, sucrose and maltodextrin. Consistent with our first prediction, nearly all these reward-specific neurons were excited by the preferred reward, maltodextrin, and most were also inhibited by the less preferred outcome, water (Fig. 5e).

Because this session was the first time rats experienced water and maltodextrin together, we were able to observe the emergence of the excitations for maltodextrin delivery, which previously produced a reduction in firing in reward-selective cells (Fig. 2k), and the emergence of inhibitions for the novel outcome, water. By averaging the normalized activity of the reward-selective cells with greater firing for maltodextrin (the same group of neurons from Fig. 5c–e), we tracked the population's responses across each trial

of each reward (Fig. 5f). In both rats, there was a noticeable increase in firing for maltodextrin and decrease in firing for water among this population of reward-selective neurons throughout the session, reflected in a significant interaction between the effects of reward and the number of trials in both rats (VP2: $F(14,1500) = 17.0$, $p = 2.1E-39$; VP5: $F(27,1960) = 22.7$, $p = 6.2E-96$). In fact, despite being classified as having greater firing for maltodextrin than water, in neither rat did these neurons start out with greater firing for maltodextrin. These data demonstrate that neurons in VP modulate their responses within minutes to reflect the relative value of available outcomes in an altered reward landscape.

**VP activity orders three outcomes by relative value.** Finally, to test whether reward-selective neurons in VP can reflect the relative value of more than two options, we conducted additional sessions for VP rats where we reintroduced sucrose along with maltodextrin and water for a total of three possible reward outcomes. We recorded activity from 254 neurons in three rats (83, 104, and 67 neurons, respectively) across four total sessions. As before, we looked for neurons with significant reward-selective responses across the three outcomes for each 600 ms bin and

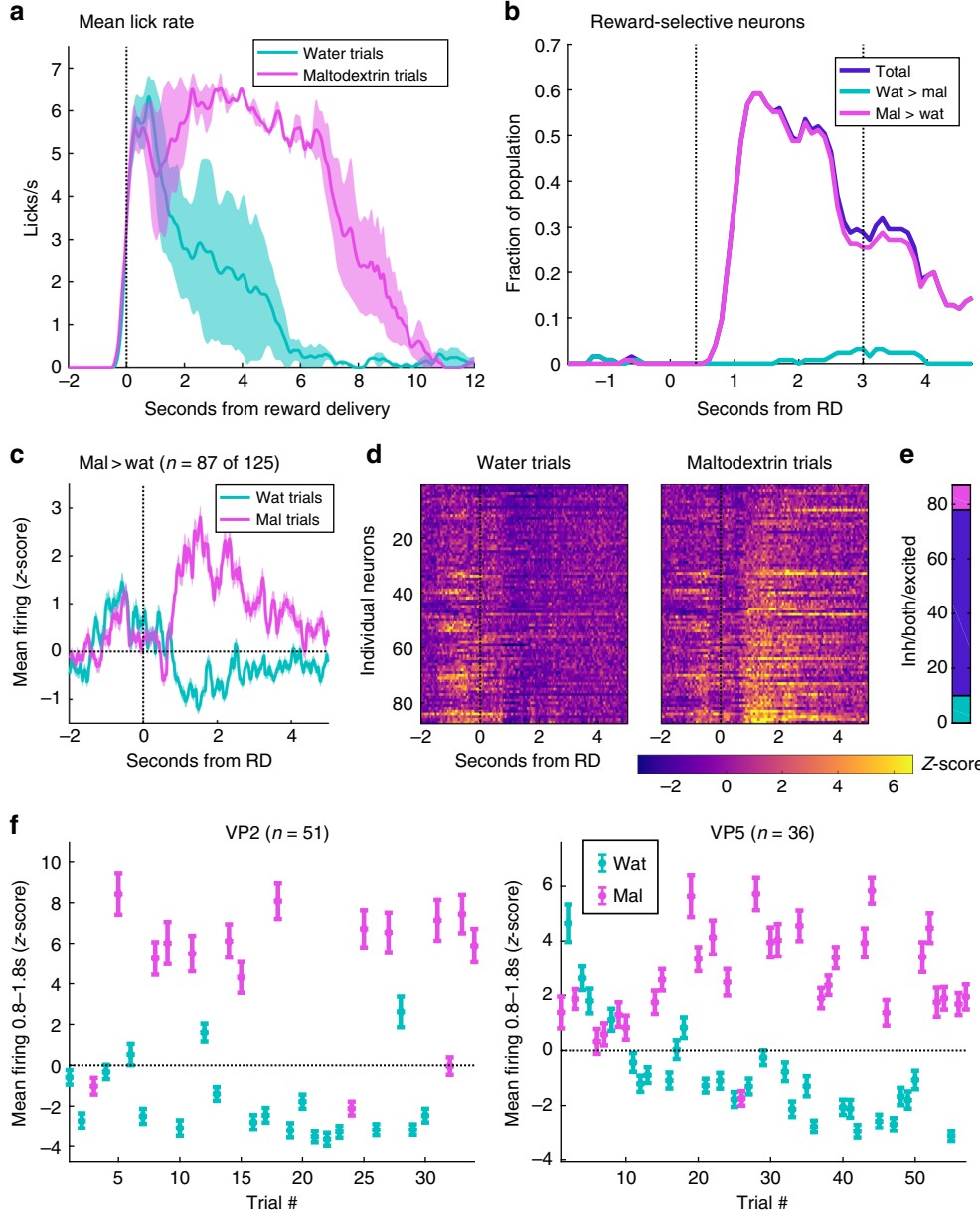

**Fig. 5** VP reward-selective activity adjusts to reflect relative value of new outcomes. **a** Lick rate on water (blue) and maltodextrin (pink) trials. Shading is SEM. **b** Fraction of VP neurons that meet criteria for reward selectivity relative to reward delivery time (as in Fig. 2). Plotted are the total fraction of reward-selective neurons (dark blue) and, of those, neurons with greater firing for maltodextrin (pink) and greater firing for water (light blue). Dashed lines indicate the window (0.4–3 s) for which reward-selective neurons were selected for (**c**) and (**d**). **c–e** Neurons with greater firing for maltodextrin in any of the bins centered 0.4–3 s. **c** Average normalized firing rate for these neurons on water (blue) and maltodextrin (pink) trials. Shading is SEM. **d** Heat maps of the normalized activity of individual neurons on water and maltodextrin trials. **e** Number of neurons with maltodextrin excitations (pink), water inhibitions (light blue), or both (dark blue). **f** Emergence of maltodextrin (pink) excitations and water (blue) inhibitions among reward-selective neurons across each completed trial of the session. Plotted as mean normalized activity 0.8–1.8 s post reward delivery; error bars are SEM

classified them by the reward that elicited the greatest firing. At most, 77% of the population was significantly modulated by reward outcome (for the bin at 1.1 s), the majority of which had greatest firing for sucrose (Fig. 6b). We then looked at the activity of reward-selective neurons with greatest firing for sucrose during any bin in our standard 0.4–3 s window. Remarkably, this population showed on average a large excitation for sucrose, a smaller excitation for maltodextrin, and an inhibition for water, consistent with the rats' relative preference for the three rewards (Fig. 6c, d). As a whole, this population had significantly different mean normalized firing rates for the time period 0.8–1.4 s after reward delivery for all three reward outcomes (F(2,561) = 441,

$p = 0.000014$; all pairs of rewards: $p < 1E-6$, Tukey test correcting for multiple comparisons). Therefore, rather than simply indicating good and bad options, VP can reliably report the relative value of multiple outcomes in a complex reward space.

## Discussion
Our data here demonstrate that neurons in both NAc and VP fire in a reward-selective manner, but this reward-specific firing is much more prevalent in the VP neural population. This relation is evident in both the larger number of neurons in VP that fire selectively for sucrose and maltodextrin, as well as the greater

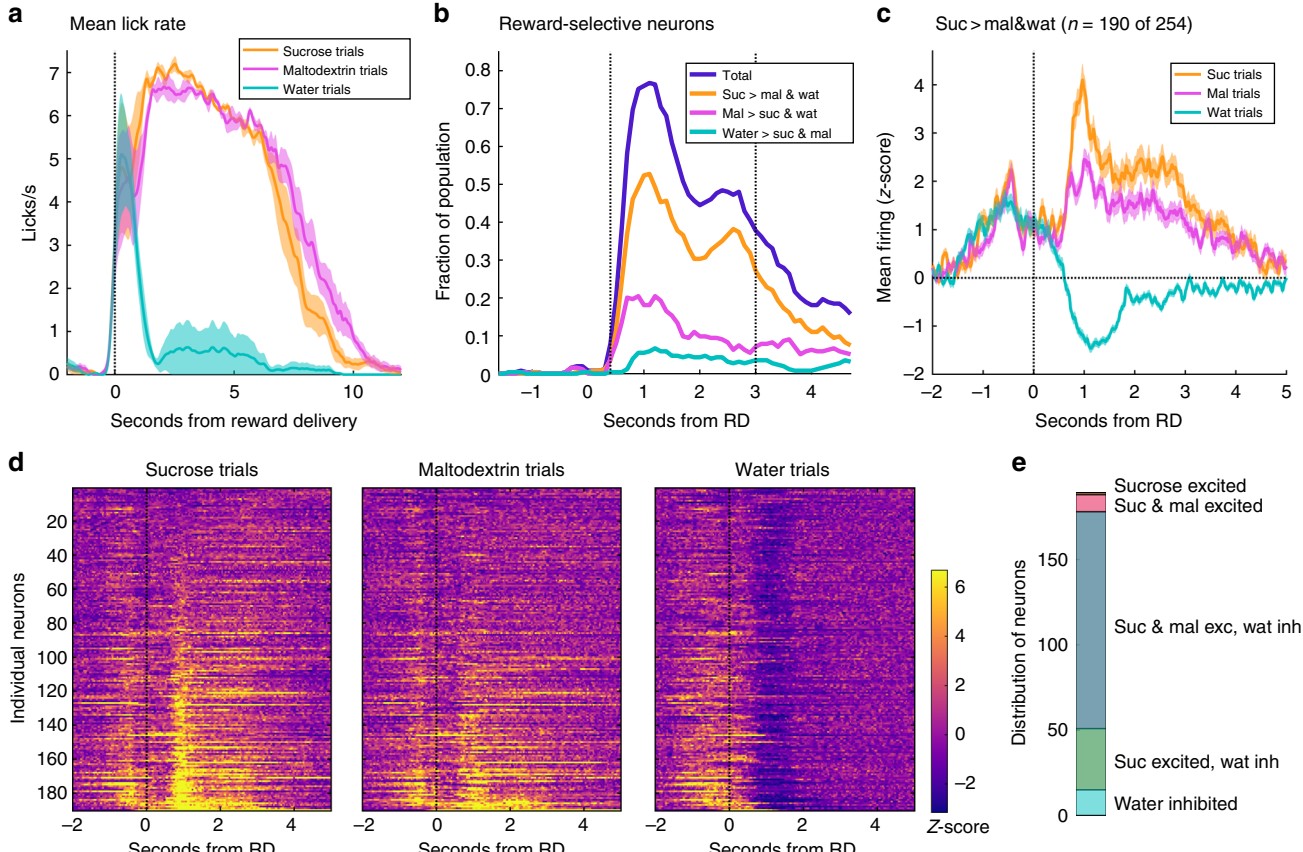

**Fig. 6** VP neurons report the relative value of three reward outcomes. **a** Lick rate on sucrose (orange), maltodextrin (pink), and water (blue) trials. Shading is SEM. **b** Histogram of the fraction of neurons in VP that meet criteria for reward selectivity relative to reward delivery time. Plotted are the total fraction of reward-selective neurons (dark blue) and, of those, neurons with greatest firing for sucrose (orange), maltodextrin (pink), or water (light blue). Dashed lines indicate the window (0.4–3 s) for which reward-selective neurons were selected for (**c–e**). **c** Mean normalized firing rate of neurons that are reward-selective for any bin 0.4–3 s post reward delivery and have greatest firing rate for sucrose. Shading is SEM. **d** Normalized firing rate of individual neurons included in **c**. Neurons in all three plots are sorted by amount of firing on sucrose trials in the bin with the most number of neurons with greatest firing for sucrose. **e** Distribution of neurons in (**c**) and (**d**) according to sucrose excitation, maltodextrin excitation, and water inhibition

decoding accuracy of LDA models trained on the spiking data of neurons in VP. Moreover, both the onset and peak of reward-specific information in VP precedes those in NAc. We also found that neurons in VP tracked the relative value of the reward outcomes across three different conditions: on a trial-by-trial basis in sessions contrasting sucrose and maltodextrin, in a new session replacing sucrose with water where maltodextrin became the preferred outcome, and, finally, in sessions with all three outcomes. Thus, our data demonstrate a robust reward valuation signal in VP that is unlikely to be fully explained by its classical NAc input.

Previous work has shown that neural responses in NAc for orally consumed rewards and their predictive stimuli are modulated by the location[44], motor response[23], size[15–17,19,42], and concentration[21,22,24] of the reward outcomes. In VP, reward-related neural responses are known to be modulated by reward size[25,26] and the rat's physiological need for a given reward[31,32]. Here, we controlled for all of these factors by choosing two reward solutions (sucrose and maltodextrin) with equivalent caloric value that were delivered in the same location and elicited nearly identical motor responses in rats in a normal physiological state. Thus, aside from their chemosensory properties, the two rewards differed only in the rats' preference for each, suggesting that the reward selectivity reported here was based on preference or identity. Because the dominant response in both regions was greater firing rate for the preferred reward, sucrose (Fig. 2e, k), it

is likely that preference is the major contributor to the reward-selective responses we observed in each region. If NAc and VP neural activity primarily coded reward identity, we would expect equivalent numbers of biased responses for each reward, along with greater rigidity of reward-specific coding in VP across changing reward contexts, two conditions that were not met. Still, the existence of a small proportion of cells in both regions with greater firing for maltodextrin (Fig. 2h, n) could be indicative of the presence of some identity-based encoding.

Due to its well-defined anatomical role as an output of NAc within the ventral striatopallidal pathway[1–3,12], most studies on the role of VP in reward processing have been within the context of NAc function. Such experiments have established an important role for this pathway in reward-related behavior. For instance, normal connectivity between NAc and VP is necessary for cues paired with reward delivery to invigorate reward-seeking actions[10]; on the other hand, disconnection of NAc and VP enhances the attribution of motivational salience to a reward-predicting cue[8], demonstrating that connections between NAc and VP have important but varying roles in the valuation of and responding to reward-related stimuli in different tasks. There is also evidence for the importance of NAc-VP connectivity in reward consumption. Normalizing plasticity between NAc D2 MSNs and their downstream targets in VP reverses deficits in hedonic responses and motivation to work for natural reward following cocaine exposure[9]; accordingly, pharmacological

inhibition of NAc D2 MSN terminals in VP increases motivation to work for food reward[45]. Additionally, NAc and VP contain reciprocally connected μ-opioid-agonist-responsive hotspots that readily alter rats' reward intake and expression of pleasure[11,46]. This collection of findings is consistent with the notion that VP is a crucial downstream mediator of NAc reward-related functions, but it does not clarify how reward-related information arrives in each region and when NAc and VP connectivity is necessary for proper reward processing, questions readily answered with in vivo observations of neural activity in each region during a reward-processing task.

A recent study of NAc and VP activity in vivo found that the onset of VP excitatory neural responses to a cue indicating reward availability typically precedes the onset of cue-evoked neural responses in NAc, demonstrating that NAc cannot be the primary source of excitatory VP cue responses and, therefore, that VP does not act exclusively downstream of NAc in the processing of cues predicting reward[28]. Likewise, in the present study, we found that reward-specific information arises and peaks earlier in VP than in NAc. This reward-specific information is largely contained within phasic excitations to the preferred reward; therefore, given that NAc inputs to VP are predominantly inhibitory (or produce a biphasic response)[12,47–50], it is unlikely that this reward-specific excitation originates in NAc. Nevertheless, our data do not exclude the possibility that certain aspects of NAc activity, such as the inhibitions we observe around the time of port entry and reward delivery (Supplementary Fig. 3b, c), are permissive of the reward-selective responses in VP, which do not arise until 0.5 s following reward delivery (Figs. 2b, 3a); additionally, later-occurring inhibitions to sucrose in VP (Fig. 2b, n, o) could originate from earlier sucrose-specific excitations in NAc (Fig. 2a, e, f). Together, our findings support the notion that VP processes certain aspects of reward independently of NAc, and they highlight the importance of studying other inputs to VP that could provide the input for the rapid, phasic reward-specific signal observed in VP here. Candidate regions include amygdala[50,51], lateral hypothalamus[52,53] and prefrontal cortex, which, in addition to direct projections[54], could provide input via the subthalamic nucleus[55,56], a route that is reported to be faster than through striatum[57,58].

In our recordings, we sampled a large proportion of the anterior-posterior extent of medial NAc shell and core and the majority of the anterior-posterior and medial-lateral axes of VP (Supplementary Fig. 1). Despite previous evidence in NAc and VP for subregion heterogeneity in reward-related function[12,13,46,59–63], we saw no meaningful differences in reward selectivity across our recorded location (Supplementary Fig. 1, Supplementary Fig. 6), which is consistent with a previous report of uniformly distributed relative value responses in NAc[21], although high density recordings in NAc and VP subregions are required to make definitive conclusions. Given the current data, our observations on the timing and magnitude of reward-selective signaling in NAc and VP appear to hold true across subregions in both structures, but the data do not preclude differences in lateral NAc shell and more rostral portions of ventrolateral VP, which we did not record from in our study, nor do they preclude different functions for a relative value signal dependent on local and long-range connectivity. Another caveat is that the neural data from each region were collected from separate animals. This approach introduces the possibility that variations in each subject's task performance and reward preference and subtle changes in the experimental conditions could contribute to the differences observed between these two groups. Future recordings performed in the same animal would provide definitive evidence that reward-specific information arises in VP prior to NAc and features more prominently in the VP neural population.

Our data show that VP neurons can flexibly signal a reward's value relative to the other currently available outcomes. A similar scheme has been shown for a small fraction of reward-selective neurons in NAc by varying the concentrations of available sucrose solutions[21,22] or the magnitude of reward[42]. While previous work has shown that VP can signal differences in value based on size[25,26], physiological need[31,32], and associative learning[40], relative value, to our knowledge, has not been tested. One noteworthy finding is that the VP neural response to heavily salinated water (normally an aversive stimulus) is greater than that of sucrose when rats are salt-deprived[31]; however, there was no significant reduction in firing for sucrose once it became the less preferred reward, perhaps because salt water and sucrose were administered in separate blocks, hindering a direct comparison. In our experiments, we have shown that the VP neural response to the same reward (maltodextrin) in the same physiological state is altered when that reward's value relative to the other available outcomes changes (Fig. 5), the hallmark of a relative value signal. The robustness of this signal across the population invites consideration of the role of ventral pallidum in the contrast effect[64]. Despite multiple demonstrations of neural correlates of negative and positive contrast in both rat and primate NAc[21,42,65,66], NAc lesions affect only instrumental but not consummatory contrast effects[67,68]; the strong relative value signal in VP makes it an appealing candidate to contribute to both effects. We also found a surprising impact of previous reward outcomes on port entry-evoked firing in VP (but not in NAc), suggesting that VP neural activity associated with reward-seeking actions reflects either an expectation of upcoming reward or a readout of recent reward history from which a relative value signal can be computed, a point of interest that could be better explored with additional studies in which expected outcome is manipulated. Overall, our findings encourage additional study of ventral pallidum function and its non-striatal inputs to better characterize its distinct role in reward processing within the ventral striatopallidal system.

## Methods

**Animals.** Subjects were male Long-Evans rats ($n = 11$) from Harlan weighing 250–275 g at arrival and single-housed on a 12 h light/dark cycle. Rats were given free access to food and water in their home cages for the duration of the experiment. All experimental procedures were performed in strict accordance with protocols approved by the Animal Care and Use Committee at Johns Hopkins University.

**Reward solutions.** Reward solutions were 10% solutions by weight of sucrose (Thermo Fisher Scientific, MA) and maltodextrin (SolCarb, Solace Nutrition, CT) in tap water. Rats were first given 24 h of free access to the maltodextrin solution in their home cages. For three subsequent days, they were given simultaneous free access to 10% solutions of sucrose and maltodextrin in their home cages.

**Behavioral task.** Rats were trained to respond to a 10 s white noise cue by making an entry into the reward port. The cue terminated upon port entry, and 500 ms following port entry, 110 μl of either reward was delivered into the metal cup within the reward port. Sucrose and maltodextrin trials were pseudorandomly interspersed throughout the session such that rats could not detect the identity of the reward until it was delivered. Individual licks were recorded with a custom-built arduino-based lickometer using a capacitance sensor (MPR121, Adafruit Industries, NY) with a 1 kHz sampling rate. Each cue was separated by a variable intertrial interval (ITI) that averaged 45 s. During the ITI, the reward cup was evacuated via vacuum pump, flushed with 110 μl of water, and evacuated again. Maltodextrin, sucrose, and water were each delivered via separate infusion pumps (Med Associates, VT) and separate metal tubes entering the cup. There were a total of 60 trials per session. In some sessions that were not included in this analysis, we presented the rewards in blocks of 30 trials each.

**Preference test.** To assay rats' preference for sucrose or maltodextrin, we performed two 60-minute two-bottle choice tests, during which rats had free access to 10% solutions of each reward. Bottles were weighed before and after to determine the amount of each solution consumed by each rat. The first test was following recovery from surgery and prior to recording. The second was at least a day after

the final session with sucrose and maltodextrin and prior to any subsequent sessions with different reward outcomes.

**Surgical procedures**. Drivable electrode arrays were prepared with custom-designed 3D-printed plastic pieces assembled with metal tubing, screws, and nuts. Sixteen insulated tungsten wires and two silver ground wires were soldered to an adapter that permitted interfacing with the headstage (Plexon Inc, TX). The drives were surgically implanted in trained rats. Rats were anesthetized with isoflurane (5%) and maintained under anesthesia for the duration of the surgery (1–2%). Rats received injections of carprofen (5 mg/kg) and cefazolin (70 mg/kg) prior to incision. Using a stereotactic arm, electrodes were aimed at either NAc (AP + 1.5 mm, ML + 1.2 mm, DV -7 mm) or VP (AP + 0.5 mm, ML + 2.4 mm ML, DV -8 mm). The base of the drive and the adapter were secured to the skull with seven screws and cement. The ground wire was wrapped around a screw and placed superficially in brain tissue in a separate craniotomy posterior to the recording electrodes.

**Recording**. Following a week of recovery in their home cages (and the first two-bottle choice test), rats were trained on the task again until they became accustomed to performing the task while tethered via a cable from their headstage to a commutator in the center of the chamber ceiling. Once they responded on at least 40 of 60 of trials, recording sessions began. Electrical signals and behavioral events were collected using the OmniPlex system (Plexon) with a 40 kHz sampling rate. We continued to record from the same location for multiple sessions if new neurons appeared on previously unrecorded channels; if multiple sessions from the same location were included in analysis, the same channel was never included more than once. If no neurons were detectable or following successful recording, the drive was advanced 160 μm, and recording resumed in the new location at minimum two days later to ensure settling of the tissue around the wires.

**Additional sessions with altered reward outcomes**. For three of the rats with electrodes in VP (VP2, VP3, and VP5), we conducted an additional session with water (replacing sucrose) and maltodextrin (VP3 did not complete the session, likely due to low motivation to pursue the new reward outcomes). The session was otherwise unchanged from those with sucrose and maltodextrin.

Subsequently, all three rats were tested in sessions with all three outcomes available. The three trial types were pseudorandomly interspersed throughout the session. The total number of trials was expanded to 90 to permit equivalent amounts of each trial as before.

**Histology**. Animals were anesthetized with pentobarbital and electrode sites were labeled by passing a DC current through each electrode. Rats were perfused intracardially with 0.9% saline following by 4% paraformaldehyde, after which brains were extracted and post-fixed in 4% paraformaldehyde for 24 h. Brains were then transferred to 25% sucrose for at minimum 24 h before being frozen on dry ice and sectioned into 50 um slices on a cryostat. Slices were then stained with cresyl violet to determine recording sites.

**Initial spike sorting and analysis**. Spikes were sorted into units using offline sorter (Plexon); following initial manual selection of units based on clustering of waveforms along the first two principal components, units were separated and refined using waveform energy and waveform heights at various times relative to threshold crossing (slices). Any units that were not detectable for the entire session were discarded. Event creation and review of individual neurons' responses were conducted in NeuroExplorer (Nex Technologies, AL). Cross-correlation was plotted for simultaneously recorded units to identify and remove any neurons that were recorded on multiple channels. All subsequent analysis was performed in MATLAB (MathWorks, MA). Event-related responses were found by constructing peristimulus time histograms (PSTHs) for spikes following each event. Neurons were determined to be modulated by an event if the spike rate in a custom window following each presentation of the event significantly differed from a 10 s window prior to cue onset according to a Wilcoxon signed-rank test ($p < 0.05$, two-tailed). For these tests, we analyzed activity 500 ms after the cue, the 1000 ms centered on port entry, and 1000 ms after reward delivery.

Optimal bin size for averaged PSTH activity was determined using Akaike Information Criteria (AIC)[14]. For our data, we used the smallest possible bin size that showed less than a 10% change from the optimal AIC value. This bin size, referred to as the deflection point, typically ranged from 20 to 100 ms. The spiking activity across these bins was smoothed with a LOWESS function.

To visualize the normalized activity of neurons, the mean activity within each of the smoothed, optimally-sized bins of the PSTH plots for each neuron was transformed to a z-score with the equation $(F_i - F_{mean})/F_{sd}$, where $F_i$ is the firing rate of the ith bin of the PSTH, and $F_{mean}$ and $F_{sd}$ are the mean and the standard deviation of the firing rate during the 10 s baseline period. Color-coded maps of individual neurons' activity and average activity traces were constructed based on these z-scores.

**Analysis of licking behavior**. PSTHs for visualizing licking activity around reward solution delivery were constructed as for neurons (above) with a fixed bin size of 100 ms and LOWESS smoothing. To test for differences in the duration of the licking bout and the number of licks on sucrose and maltodextrin trials, we ran a three-way ANOVA on the raw licking data for the fixed effect of reward and the random effects of session and subject, with session nested within subject (with trial as our n). We also ran this test on the number of licks 1–4.5 s post reward delivery, an epoch in which we noticed a visible difference in the average lick rate (Fig. 1d). We further characterized this difference in licking activity by finding the mean duration of the interlick intervals following the first 30 licks of each reward. We ran a three-way ANOVA for the fixed effects of reward and interval # and the random effect of subject (with each ILI's session mean as our n).

**Classification of neurons as reward-selective**. For the analysis of reward-selective activity during reward consumption, we segmented the time surrounding reward delivery into overlapping 600 ms bins advanced by 100 ms. We only included trials in which the rat began licking within 2 s of reward delivery to ensure the rat sampled the reward on each included trial. Neurons were significantly reward-modulated for a given bin if they there was a significant interaction ($p < 0.01$) for that neuron between the effect of baseline ($-22$ to $-12$ s from reward delivery) vs. bin firing and the effect of reward solution (with trial as our n) in two consecutive bins. This approach minimized the amount of noise in the classification (measurable as the number of neurons classified as reward selective prior to reward delivery) while still permitting relatively brief reward-specific responses to register. We then further classified these reward-selective neurons by the reward for which they had greater normalized firing in that bin, found with the equation $(F_b - F_{mean})/F_{sd}$, where $F_b$ is the firing rate of each bin, and $F_{mean}$ and $F_{sd}$ are the mean and the standard deviation of the firing rate during the 10 s baseline period. This same analysis was used to classify neurons from the sessions comparing water and maltodextrin.

To choose which bins best captured the population of reward-selective neurons across both regions, we plotted the cumulative onset of reward selectivity for all neurons as a fraction of the total population (Fig. 2a, b). We chose to include all neurons that were reward-selective in any of the bins from 0.4 to 3 s, which captured the majority of phasic reward-specific responses following reward delivery in both regions. To determine which of these neurons were significantly excited or otherwise inhibited by either reward (Fig. 2g, j, m, p), we performed a Wilcoxon signed-rank test comparing on each trial the (raw) firing rate during the $-22$ to $-12$ s baseline window from reward delivery to the firing rate in each of the bins centered 0.4–3 s for each reward ($p < 0.05$ cutoff, two-tailed). A neuron was considered excited or inhibited by a given reward if it had a significant increase or decrease in spikes for any of the bins 0.4–3 s post reward delivery. We also plotted the cumulative onsets of reward-selective neurons as a fraction of total reward selective neurons in each region to compare the timing of the onsets in each region and compared the distributions with a two-way ANOVA with the main effect of region and the random effect of subject (Fig. 2d).

To classify neurons as reward-selective with three reward outcomes, we performed the same ANOVA analysis as before with the water condition added to the effect of reward, looking for an interaction between the effects of reward and baseline vs. bin firing (with trial as our n). We then further classified reward-selective neurons by the outcome for which they had the greatest spiking in that bin and found, as before, if a neuron was significantly inhibited or excited by any of the outcomes in any bin 0.4–3 s with Wilcoxon signed-rank tests ($p < 0.05$ cutoff, two-tailed). Because there were three outcomes, we also performed a two-way ANOVA on the effect of reward outcome (and random effect of subject) on the average normalized firing 0.8–1.4 s post reward delivery (the bin with the most number of reward-selective neurons) of all selective neurons with greatest firing for sucrose (Fig. 6c) as well as pairwise comparisons between the three rewards (Tukey test, correcting for multiple comparisons).

**Quantification of firing rate based on current and previous reward**. To examine how average activity in each region was affected by previous reward, we normalized the average activity of all neurons in each region to their baseline firing rate in a window $-22$ to $-12$ s from reward delivery. We chose to quantify the average activity 0.8–1.3 s post reward delivery (marked with blue lines in Fig. 4a, c), a period we visually identified as having the best evidence of previous reward-modulated activity. Thus, the activity of neurons was normalized with the equation $(F_r - F_{mean})/F_{sd}$, where $F_r$ is the mean firing rate 0.8–1.3 s following reward delivery for each of the four current/previous reward combination, and $F_{mean}$ and $F_{sd}$ are the mean and the standard deviation of the firing rate during the 10 s baseline period on all trials. We then performed ANOVAs testing the effects of reward and previous reward (and random effect of subject) on the normalized activity of neurons in that window for each region (with neuron as our n). To compare the regions, we also performed a test on all the neurons from both regions with the added factor of region.

**Linear models**. To find the impact of previous trials' outcomes on current trial firing, we fit linear models (fitlm in MATLAB) to the firing rate of each neuron on each trial according to the outcomes on the current trial and the previous six trials. For this analysis, we used the same window as above, 0.8–1.3 s post reward delivery, and normalized the activity for each neuron on each trial to the activity of

that neuron during baseline period −22 to −12 s from reward delivery on all trials. The normalized activity on each trial was paired with a corresponding vector of seven 0 s and 1 s indicating the reward outcome (0 for maltodextrin and 1 for sucrose) on the current and previous six trials (this required exclusion of all trials preceding the seventh completed trial). This convention caused positive coefficients to indicate a positive influence of receiving sucrose rather than maltodextrin on firing rate for that trial and vice versa. We then found the coefficients for each of the seven relative trials for each neuron as well as whether there was a significant impact of that relative trial on firing rate ($p < 0.05$, two-tailed $t$-test). We then did the same analysis but shuffled the trial outcomes to find what values would be expected by chance. For each region, we performed ANOVAs testing the main effects of shuffled vs. true data and trial relative to current (and the random effect of subject) on coefficient and then performed Tukey tests correcting for multiple comparisons to find differences on each trial between the coefficients and their shuffled data ($p < 0.05$). We tested for significant differences in the proportion of neurons with significant coefficients across true and shuffled data from both regions as well as just the true data from each region with chi-squared tests for each relative trial ($p < 0.05$). For the PE coefficients in VP, we also performed an ANOVA testing for the main effect of whether a neuron was classified as selective in Fig. 2 on the strength of the coefficients in the first two previous trials (in addition to the effect of trial and random effect of subject).

**Emergence of responses to water and maltodextrin.** To track how the average activity of reward-selective neurons changed on water and maltodextrin trials across the session with those two reward outcomes, we normalized the mean activity of the reward-selective neurons identified in Fig. 5c–e on each trial to their baseline activity in the 10 s window −22 to −12 s from reward delivery. We focused our analysis on each neuron's normalized activity 0.8–1.8 s following reward delivery, an epoch we visually identified as representative of the maltodextrin excitations and water inhibitions. Thus, neurons were normalized with the equation $(F_t − F_{mean})/F_{sd}$, where $F_t$ is the mean firing rate 0.8–1.8 s following reward delivery on each trial, and $F_{mean}$ and $F_{sd}$ are the mean and the standard deviation of the firing rate during the 10 s baseline period on all trials. We then plotted the average activity according the number of trials the rat had completed (Fig. 5f). We performed a two-way ANOVA (reward X trials of reward) on the normalized activity of the neurons from each rat across each respective trial of each reward (with each neuron's normalized activity on each trial as our $n$). This approach required capping the total number of trials included in the test at the maximum number of trials for the reward with the least number of completed trials.

**Decoding.** For single unit decoding, a linear discriminant analysis (LDA) model (the fitcdiscr function in MATLAB) was trained on one neuron's spike activity for one 600 ms bin on 80% of trials. This model was then used to classify the remaining 20% of trials as sucrose or maltodextrin in a fivefold cross-validation approach and averaged performance across all five repetitions to find that unit's accuracy. We also conducted the analysis with the trial identities shuffled to determine the accuracy on shuffled data. We then repeated this analysis for every neuron in each region for each bin. If there were fewer than seven spikes across all sucrose or maltodextrin trials, we excluded that neuron for that bin to avoid errors from creating an LDA model on a dataset with too little variance. To determine when accuracy in each region improved over shuffled data, we found all bins when the mean accuracy of the true data exceeded the 99% confidence interval of the shuffled data for at minimum two consecutive bins. To ensure that our results were not affected by the greater number of neurons in VP (423 vs. 182), we took 20 random selections of 182 of the unit models from VP and recalculated the confidence intervals to evaluate if it would affect the results (by and large it did not; see Results). To compare accuracy in our standard window of 0.4–3 s after reward delivery (Fig. 2) across regions, we performed an ANOVA testing the effects of shuffled versus true data (whether or not the accuracy came from a shuffled data model or a true data model), region, and bins (and the random effect of subject) with each neuron model's true or shuffled accuracy in each bin as our $n$. We also performed an ANOVA testing the effects of shuffled vs. true data and region (and the random effect of subject) to compare the accuracy of the most accurate bin in each region (with the shuffled and true data from each neuron in the respective bin from each region as our $n$). To compare only reward-selective neurons (Supplementary Fig. 7), we performed the same tests but included only neurons classified as reward-selective in Fig. 2.

To look at how model classification accuracy increased with additional units, we pooled together separately recorded units. This approach requires matched numbers of trials, so we only included neurons recorded during sessions with at least 20 trials of each reward. Subsequently, when training our pseudoensemble LDA models, we restricted the analysis to 20 (randomly selected) trials of each reward. We found the fivefold cross-validated accuracy for models trained on the activity of randomly selected levels of 10, 25, 50, 100, and 150 units from each region. For each level, we performed the analysis 50 times. We then performed a two-way ANOVA on the effects of pseudoensemble size and region on the accuracy

at each level's peak bin (with each repetition at that peak bin for each level as our $n$). We also performed a two-way ANOVA on the effects of pseudoensemble size and region on the time of most accurate bin for each LDA model replicate (with each repetition's peak bin time at each level as our $n$). We also performed these analyses on pseudoensembles containing only reward-selective neurons as classified in Fig. 2 (Supplementary Fig. 7).

**Statistical analysis.** Data are presented as mean ± s.e.m. unless otherwise noted. Statistical analyses were performed in MATLAB (MathWorks) on unsmoothed data. Specific tests are noted in the text, figure legends, and throughout the methods. Generally, we used analysis of variance (ANOVA) tests (the anovan function in MATLAB) to test for main effects and interactions, Tukey tests for pairwise comparisons corrected for multiple comparisons (multcompare in MATLAB), and chi-squared tests for contingencies (crosstab in MATLAB). For all ANOVAs testing behavioral and neural data across subjects, we included the random effect of subject to account for non-independence in the data.

## Data availability

The data that support the findings of this study and the code used to analyze and visualize the data are available in an online repository [https://doi.org/10.12751/g-node.b6d000] (ref. [69]).

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

## Acknowledgements

This work was supported by National Institutes of Health grants 5 T32 NS91018-17 (D. O.), K99 AA025384 (J.M.R.), and R01 DA035943 (P.H.J.), by a NARSAD Young Investigator Award (J.M.R.), and by the National Science Foundation Graduate Research Fellowship under Grant No. DGE-1746891 (D.O.). The authors thank James Garmon for assistance with equipment design and implementation.

## Author contributions

D.O., J.M.R. and P.H.J. designed the experiments; D.O. collected, analyzed, and visualized the data; D.O., J.M.R. and P.H.J. interpreted the data and wrote the manuscript.

## Additional information

**Competing interests:** The authors declare no competing interests.

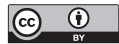

