## [Peer Review File · Nature Communications]

Reviewers' comments:

Reviewer #1 (Remarks to the Author):

Ottenheimer and colleagues examined whether the ventral basal ganglia, which contain the nucleus accumbens (NAc) and ventral pallidum (VP), can encode the relative reward value during performing reward-oriented actions. The authors demonstrated that the reward-associated responses in VP occurred earlier than those in NAc, and concluded that the reward information in VP does not come from NAc. They also showed that neurons in VP tracked the relative value of delivered reward.

Overall, the data acquisition is technically solid, and the manuscript is interesting and well written. However, I have several concerns before the publication of this study, which can deepen our understanding the functions of ventral basal ganglia.

(Major concerns)

1. In my view, one of main strengths of this study is the quantitative demonstration of neuronal comparison between NAc and VP using the same task. Based on the onset latencies of reward-associated responses between NAc and VP, the authors concluded that the origin of reward information in VP is not the NAc. This conclusion, however, is contrary to the traditional concept regarding the information flow in the ventral striatopallidal axis. For example, since the striatal input to VP is inhibitory, the sucrose-preferred responses in NAc neurons can induce the maltodextrin-preferred responses in some VP neurons. If the authors like to keep their conclusion, I would like to see the data that the VP responses will not disappear after the inactivation of NAc (using the method of optogenetics or classic muscimol injection). Otherwise (if these experiments are technically difficult), the authors should weaken the conclusion and elaborate the discussion.

2. Even if the common inputs to NAc and VP are important for reward information in both regions, the authors cannot deny the information flow in the ventral striatopallidal axis. In particular, I am interested whether NAc can be the source of relative value signals in VP. If possible, I suggest the authors to show the data if NAc neurons can encode the relative value of three reward outcomes.

3. The author claimed that the analysis of reward history revealed the relative value signaled in NAc and VP (Fig. 4). Some previous studies have indicated that neurons in the ventral basal ganglia signal the expectation of reward. If this is the case, there may be a possibility that cue responses in NAc and VP changed depending on the reward history (Supplementary Fig. 3). For example, the probability of sucrose trial would be higher after maltodextrin trial due to the pseudorandom trial design. I suggest the authors to analyze this effect.

I am also interested whether the ventral basal ganglia can signal the reward prediction error. In this respect, I would like to know whether there exist some NAc and VP neurons signaling the reward prediction error.

4. In Fig 6d, most of the sucrose-preferred neurons showed strong inhibition in water trials. If VP neurons signal the relative reward value, the inhibitory response should be observed in maltodextrin trials in the task with two reward options. However, the inhibitory response in VP was weaker (Fig. 2i). How do the authors explain?

(Minor comments)

1. Fig. 5c should be divided into Fig. 5c, d, e (the same format as in the other figures).

2. In Methods, please describe the sampling rates of licking measurement and neuronal recording.

Reviewer #2 (Remarks to the Author):

Ottenheimer et al compare firing responses in the ventral pallidum (VP) and a nucleus accumbens (NAc) during consumption of different liquid rewards. They find that VP neurons' firing is robustly related to the rat's relative preference, whereas NAc neurons' firing is less influenced by preference. In addition, reward-specific firing occurs earlier in the VP than the NAc. Finally, VP neurons' reward firing is strongly related to the other rewards available during the task, such that rewards that elicit no excitation when more preferred rewards are also available come to evoke excitation when the other reward is less preferred. This work is of high theoretical significance because it strongly implicates the VP in reward-related functions that are often ascribed (based on little direct evidence) to the NAc – and further suggests that the relevant VP encoding is at least partially independent of the NAc. The electrophysiological and behavioral aspects of the study are of high quality and the results are convincing. However, I have some suggestions for further analyses that would clarify some of the points raised in their manuscript.

1. It is not immediately obvious from any of the graphs in Fig. 2 that the first signs of differential firing for the two rewards occurs first in the VP. I suggest adding a graph that explicitly compares the fraction of reward-selective VP and NAc neurons across time. For instance, the authors could subtract the two fractions (or fractional values normalized to the maximal fraction that is selective in each structure) and plot the difference across time.

2. The authors conclude that reward-selective firing in the NAc is unlikely to account for the more robust selectivity in the VP. However, this leaves open the question of whether the VP activity that encodes rewards is nevertheless dependent on activity in the NAc. For instance, according to fig. S3, about a quarter of NAc neurons are inhibited after reward delivery whereas nearly half of VP neurons are excited. Could it be that the NAc inhibitions result in disinhibition that is essential for the VP excitations, even if the reward selectivity of the VP excitations is driven by other projections to the VP? One way to shed light on this sort of question is to ask when changes in firing rate occur in each structure: do reward delivery-elicited inhibitions in NAc reliably precede excitations in VP? And the same for excitations in NAc and inhibitions in VP? Admittedly, this analysis may not be conclusive, and it is likely to be complicated by overlap from the port entry-related changes in firing. At the very least, in the Discussion the authors could make the point that the fact that VP encoding of reward precedes that in the NAc does not imply that the VP encoding does not somehow depend on input from the NAc.

3. The authors report that the degree of reward selectivity in VP firing depends on the previous reward. This is a fascinating result that leads to several questions the authors could easily address with the existing data:

A. The analysis asked, very simply, if the reward on the previous trial influenced selectivity on the current trial. However, it's possible that the neurons are influenced by trials occurring before the previous trial. The authors could examine what the "look-back period" is that influences current-trial reward encoding, as well as determine the degree to which previous trials influence the present trial. Bayer and Glimcher (Neuron 2005) give an idea of how to do this using regression (e.g., their fig. 5). This question is important because the authors' data indicates that reward context is important for defining VP neurons' relative reward value encoding – but "context" can mean many things, and this analysis would help to constrain the factors that define the way in which VP neurons' reward selectivity is relative. (Notably, fig. 5d suggests that at least 5 trials, and maybe more, are required to "flip" VP neurons' firing after the context changes. That suggests that the look-back period may be

considerably longer than 1 trial.)

B. The analysis described in A brings to mind the question of whether a recent history effect would be stronger in NAc neurons if more past trials were included. For instance, suppose the animal obtains 4 maltodextrin rewards in a row followed by a sucrose reward. Would the reward response of NAc neurons to the sucrose reward be different than on sucrose trials preceded by 4 sucrose rewards? Looking just one trial back might be insufficient to pull out such an effect.

C. Once the authors have established the recency parameters that influence selectivity of reward delivery-evoked firing, the next question is whether there is a recency effect in any other form of neural activity observed in this task – for instance, firing evoked by cue presentation and port entry.

D. The reward context-dependent firing could play a role in positive and negative contrast behavioral experiments. It would be useful for the authors to discuss such experiments and whether a potential role for the VP in contrast effects has been explored.

E. Recency effects are not described for the data in fig. 6 (delivery of 3 different rewards) even though the data set appears to be rich enough to support such an analysis.

4. The authors should discuss the caveat that VP and NAc recordings were made in different animals and therefore in different sessions, and describe what might be gained from making the recordings in the same animals.

5. I suggest the authors change the title of the paper to reflect the differences between NAc and VP.

Reviewer #3 (Remarks to the Author):

In this manuscript by Ottenheimer et al, the authors record from the nucleus accumbens (NAc) and ventral pallidum (VP) during a task in which a single cue predicts two differently preferred outcomes with identical caloric value. Two separate groups of rats (n = 6 NAc and n=5 VP) were implanted with drivable recording electrodes and authors examine single-unit encoding of these two liquid reinforcers, sucrose and maltodextrin across multiple sessions. In the first phase of the experiment, authors observe substantial proportions of neurons (40% NAc and 63% VP) that they classify as reward-responsive in the two regions. The authors report that a larger proportion of NAc and VP neurons prefer sucrose to maltodextrin than vice versa, similar to the homecage preference displayed by majority of animal prior to recording. The authors use a decoder strategy to use observed NAc or VP spiking to predict what outcome was delivered on a subset of trials, and report that both NAc and VP neurons accurately decode reward identity. The present a case that VP decoding accuracy is better than NAc decoder accuracy. Using observed neural activity and decoding accuracy the authors also present evidence that VP encoding of reward identity precedes that of NAc. The authors then investigate the possibility that NAc and VP may encode relative reward by first examining reward responses relative to the identity of reward on prior trial. With evidence that VP, but not NAc, neurons respond differently depending on the identity of reward on the prior trial, the authors use a subset of rats to examine relative encoding of the less preferred outcome, maltodextrin versus water, observing greater neural preference for maltodextrin (excitation) compared to water (inhibition). When all three reward are made available, the relative value encoding (suc>malt>water) predicted from prior observations holds true.

This manuscript is well written and relevant background is presented to set up the importance for this study to resolve inconsistent reports about the serial vs parallel reward processing in NAc and VP.

Generally, the results are quite intriguing, and the use of a decoder strategy to better understand the contribution of NAc and VP encoding to reward identification is commendable. There are several questions/concerns about statistical analyses and the implementation of the decoder that arise in methods/results sections. If these issues can be addressed and the results hold true, this will be an interesting set of findings that will make a substantial contribution to our understanding of NAc and VP reward encoding.

The first major point is that the degrees of freedom of the error in ANOVA analyses are very high (number of observations in analysis: ~ 3300 for lick data (from 63 sessions), ~ 2600 for neural activity, and $\sim 81,000$ for decoder accuracy). For both lick and neural data analyses, can the authors clarify whether they are using bin, trial, or session as their n ? Using multiple bins (per trial across many trials per session) as the number of observations in the analysis would be problematic in that it inappropriately inflates the number of observations and statistical power. It is also unclear whether repeated measures are being accounted for in the analyses presented. A related point applies for z -score data (lines 647-652). For the z score calculation each bin of interest (variable "optimal" ~ 20 -100 ms) is compared to a 10 s pre-cue baseline period. Please clarify: are the F_{mean} and F_{sd} for the baseline period calculated where n is trial (ie many 20-100 ms bins averaged across 10 s to have a per trial value) or are these calculated where n is bin, thus many hundreds of bins inflate number of observations going into the baseline mean/sd calculations? Please report if there is a correction for multiple comparisons.

The second major point is that the decoder approach does not take into account different proportions of reward responsive neurons in NAc and VP, and thus the conclusions drawn from this powerful strategy, may not reflect real regional differences in reward responses in NAc and VP. The authors report that a significantly larger proportion of VP neurons (63%) are reward responsive compared to NAc (40%). Further, in peak analyses (for which much of the decoder approach focuses) the authors report 37% VP vs 14% NAc. Thus, a strategy that uses random selection of neurons across the entire population is more likely to randomly select a reward responsive VP neuron than a reward responsive NAc neuron. Therefore, the VP population may appear to be better able to decode reward identity. The authors could instead use only the reward-related neurons to do the decoding, or if this strategy doesn't work due to different numbers of neurons recorded, the authors could employ a normalization strategy, such that the likelihood of randomly selecting a reward responsive neuron is similar for VP and NAc. Alternatively, (though less ideally) authors could do a post-hoc analysis of the presented decoder analyses showing no difference in number of reward selective neurons randomly represented in VP and NAc simulations, to build a case that the data presented is evidence of superior VP decoder accuracy.

Minor:

Could the authors please clarify whether the stats done on raw or smoothed data?

Fig 3: Shuffled VP or NAc and why get to $\sim 70\%$ accuracy? For peak analyses, how did authors determine peak bin for shuffled data? Is it 1.0 s for VP and 1.7 s for NAc?

Fig 3 Is Shuffled vs true a subtraction score? Bin minus relative bin?

Fig 4 stats ANOVA on Z -score or raw data? What is sig. z -score value?

Fig 5 line 244 (text) comparison is unfair to make due to only a subset of rats being used. A better comparison would be to compare proportion of reward responsive in these two rats across two dif types of reward sessions.

Could not find stats for 5D.

Could not find statistical analyses for figure 6 data

Y-axis in Fig. 5A is not labeled.

Sup fig 5 is an excellent resource.

Response to reviewers

Reviewer 1:

We are pleased that Reviewer 1 found our manuscript “technically solid,” “interesting,” and “well written.” The reviewer brought up multiple concerns that we were happy to address here and in the manuscript.

Major points

“1. In my view, one of main strengths of this study is the quantitative demonstration of neuronal comparison between NAc and VP using the same task. Based on the onset latencies of reward-associated responses between NAc and VP, the authors concluded that the origin of reward information in VP is not the NAc. This conclusion, however, is contrary to the traditional concept regarding the information flow in the ventral striatopallidal axis. For example, since the striatal input to VP is inhibitory, the sucrose-preferred responses in NAc neurons can induce the maltodextrin-preferred responses in some VP neurons. If the authors like to keep their conclusion, I would like to see the data that the VP responses will not disappear after the inactivation of NAc (using the method of optogenetics or classic muscimol injection). Otherwise (if these experiments are technically difficult), the authors should weaken the conclusion and elaborate the discussion.”

Response: The reviewer raises a good point that the current study does not rule out a role for NAc in providing some of the reward-selective responses in VP. We have a few concerns with attempting to inactivate NAc while recording in VP in order to address this point. The reviewer points out that the experiment is technically difficult: in addition to the added difficulty of successfully implementing two techniques in the same animal, the primary issue is that NAc and VP are adjacent regions, so any inactivation of NAc would have the potential to inadvertently inactivate part of VP as well. Ultimately, we feel that the data collected from such an experiment would be hard to interpret. If inactivating NAc has no effect on the VP reward-selective signal, we might worry that the inactivation did not fully affect all of the NAc inputs to VP. If inactivating NAc does reduce reward-selective signaling in VP, it would be hard to determine whether NAc is actually the source of the reward-selective signaling or whether it is simply permissive of normal VP activity. So, to address the reviewer's concern, we have taken his/her advice, and have revisited each claim we make about the relationship between NAc and VP activity to allow for the possibility that NAc is involved in some of VP reward-selective firing, and we discuss this point more in the discussion.

2. Even if the common inputs to NAc and VP are important for reward information in both regions, the authors cannot deny the information flow in the ventral striatopallidal axis. In particular, I am interested whether NAc can be the source of relative value signals in VP. If possible, I suggest the authors to show the data if NAc neurons can encode the relative value of three reward outcomes.

Response: We agree that we cannot deny the literature demonstrating the importance of NAc inputs to VP in a variety of behaviors. Therefore, to address the reviewer's concern, we have

modified the text to add discussion on the possibility that NAc responses at the time of the port entry or reward delivery may be permissive of VP reward-selective firing. As for NAc having the ability to encode the relative value of three outcomes, we feel that work from the Fields lab (Taha et al, 2005) addresses this point: they find a modest number of neurons with graded firing according to the concentration of sucrose, so while we do believe that some neurons in NAc can report relative value, we suspect should we perform this experiment that NAc neurons would continue to do so more slowly and less robustly than VP neurons, and thus we do not believe it would contribute additional insight to our findings.

3. The author claimed that the analysis of reward history revealed the relative value signaled in NAc and VP (Fig. 4). Some previous studies have indicated that neurons in the ventral basal ganglia signal the expectation of reward. If this is the case, there may be a possibility that cue responses in NAc and VP changed depending on the reward history (Supplementary Fig. 4). For example, the probability of sucrose trial would be higher after maltodextrin trial due to the pseudorandom trial design. I suggest the authors to analyze this effect.

I am also interested whether the ventral basal ganglia can signal the reward prediction error. In this respect, I would like to know whether there exist some NAc and VP neurons signaling the reward prediction error.

Response: We appreciate the suggestion to analyze our data for these effects. In response to this point and to the points raised by Reviewer 2, we have more thoroughly analyzed the impact of reward history on cue- and port entry-evoked firing, which we have summarized in Figure S7. Interestingly, there is no impact of previous outcome on cue-evoked firing, but there is an effect on port entry-evoked firing in VP (which is stronger in reward-selective neurons), so perhaps the port entry response in VP reflects, in part, an expectation of reward, or at least the recent reward history, a point we have added to the discussion. We have also attached the new supplementary figure in this letter in response to Reviewer 2's comment.

As for the presence of reward prediction error (RPE) in each region, this is an important issue. We feel that a rigorous test of RPE, which we would implement with multiple cues of varying value, but a predictable outcome, as well as reward omissions, is beyond the scope of our experiment. We did, however, look for the number of neurons in each region with cue excitations and then reward-selective responses, which is how we would expect an RPE-neuron to respond in our task. We found that 10% of neurons in NAc (42% of the reward-selective population) and 29% of neurons in VP (56% of the reward-selective population) fire in this manner, and we have provided those results here. To further explore the intriguing possibility that VP encodes RPE, we are currently planning additional studies with multiple cue-outcome contingencies, the results of which will be beyond the scope of this manuscript.

Response Letter Figure 1. Reward prediction error-like responses in NAc and VP. Here, we plot neurons in NAc and VP with both significant excitations in response to the and reward-selective responses following reward delivery, as would be expected by a neuron representing a reward prediction error. Such neurons constituted 10% of the neural population in NAc and 29% of the population in VP.

4. In Fig 6d, most of the sucrose-preferred neurons showed strong inhibition in water trials. If VP neurons signal the relative reward value, the inhibitory response should be observed in maltodextrin trials in the task with two reward options. However, the inhibitory response in VP was weaker (Fig. 2i). How do the authors explain?

Response: We note that the degree of difference between maltodextrin and water is much greater than maltodextrin and sucrose, as evidenced behaviorally by the nearly identical licking patterns for sucrose and maltodextrin (Fig. 1d) but very little licking for water (Fig. 5a). The fact that sucrose and maltodextrin are both appetitive stimuli while water is not may explain why the inhibitions are not as strong. We believe that a relative value signal need not have the same minimum report of value in all scenarios; the chief element of a relative value signal is an adjustment of the report of a stimulus's value relative to other available outcomes (Fig. 5f). To address the reviewer's point and clarify this for the reader, we have made note of this point in the manuscript where we describe the result.

Minor points

1. *Fig. 5c should be divided into Fig. 5c, d, e (the same format as in the other figures).*

Response: We have added these additional panel divisions, thank you.

2. *In Methods, please describe the sampling rates of licking measurement and neuronal recording.*

Response: We have added this information to the methods. OmniPlex (our neuronal recording system) samples at 40kHz, and the capacitance sensor samples at 1kHz.

Reviewer 2:

We are glad that Reviewer 2 saw our manuscript as “high quality” and found the results “convincing.” We are especially happy that the reviewer agreed with us that “this work is of high theoretical significance because it **strongly implicates the VP in reward-related functions that are often ascribed (based on little direct evidence) to the NAc** – and further suggests that the **relevant VP encoding is at least partially independent of the NAc.**” They provided a number of suggestions for data analysis that we have implemented, as detailed below.

1. *It is not immediately obvious from any of the graphs in Fig. 2 that the first signs of differential firing for the two rewards occurs first in the VP. I suggest adding a graph that explicitly compares the fraction of reward-selective VP and NAc neurons across time. For instance, the authors could subtract the two fractions (or fractional values normalized to the maximal fraction that is selective in each structure) and plot the difference across time.*

Response: We have implemented this helpful suggestion by adding a subtraction comparison in Fig 2c (and also provided below), which now reveals that at no point are there more reward-specific responses in NAc than in VP, a point we have also added to the text. We also plotted the cumulative onset of reward-selective responses as a fraction of total reward-selective responses to reveal the difference in time course of reward selectivity, which reveals a significantly earlier distribution in VP.

Figure 2. More neurons in VP fire selectively for sucrose and maltodextrin than in NAc.

(a,b) Top panel: fraction of NAc (a) and VP (b) neurons meeting criteria for reward selectivity as a function of time after reward delivery. Plotted are total fraction of reward-selective neurons (blue) and, of those, neurons with greater firing for sucrose (orange) and greater firing for maltodextrin (pink). Bottom panel: Cumulative distribution of reward selectivity over time after reward delivery. (c) Subtraction of VP reward selectivity from NAc in each bin. Negative values indicate more selectivity in VP. (d) Cumulative distribution of reward selectivity onsets as a fraction of total reward-selective neurons. Asterisk indicates significantly earlier onsets in VP ($F(1,290) = 12.7, p = 0.00071$).

2. The authors conclude that reward-selective firing in the NAc is unlikely to account for the more robust selectivity in the VP. However, this leaves open the question of whether the VP activity that encodes rewards is nevertheless dependent on activity in the NAc. For instance, according to fig. S3, about a quarter of NAc neurons are inhibited after reward delivery whereas nearly half of VP neurons are excited. Could it be that the NAc inhibitions result in disinhibition that is essential for the VP excitations, even if the reward selectivity of the VP excitations is driven by other projections to the VP? One way to shed light on this sort of question is to ask when changes in firing rate occur in each structure: do reward delivery-elicited inhibitions in NAc reliably precede excitations in VP? And the same for excitations in NAc and inhibitions in VP? Admittedly, this analysis may not be conclusive, and it is likely to be complicated by overlap from the port entry-related changes in firing. At the very least, in the Discussion the authors could make the point that the fact that VP encoding of reward precedes that in the NAc does not imply that the VP encoding does not somehow depend on input from the NAc.

Response: We thank the reviewer for raising this important point. We agree (and suppose that it is likely) that activity in NAc is permissive of certain aspects of VP firing. As the reviewer points out, it would be difficult to determine whether reward-elicited excitations in VP reliably follow reward-elicited inhibitions in NAc because of the close proximity of the PE, which also elicits responses in a majority of the population of each region. Nevertheless, it is clear that there are many inhibitions in NAc whose onset precedes reward delivery (Fig S2c), and the reward-specific signal in VP first arises around 0.4s following reward delivery (Fig 3a). Thus, we have

added to the discussion the important point the reviewer makes: that proper VP reward-selective firing may depend on NAc input even if NAc is not the likely source of the reward-specific information.

3. The authors report that the degree of reward selectivity in VP firing depends on the previous reward. This is a fascinating result that leads to several questions the authors could easily address with the existing data:

A. The analysis asked, very simply, if the reward on the previous trial influenced selectivity on the current trial. However, it's possible that the neurons are influenced by trials occurring before the previous trial. The authors could examine what the "look-back period" is that influences current-trial reward encoding, as well as determine the degree to which previous trials influence the present trial. Bayer and Glimcher (Neuron 2005) give an idea of how to do this using regression (e.g., their fig. 5). This question is important because the authors' data indicates that reward context is important for defining VP neurons' relative reward value encoding – but "context" can mean many things, and this analysis would help to constrain the factors that define the way in which VP neurons' reward selectivity is relative. (Notably, fig. 5d suggests that at least 5 trials, and maybe more, are required to "flip" VP neurons' firing after the context changes. That suggests that the look-back period may be considerably longer than 1 trial.)

Response: We found the possibility that trials beyond the most recent affect NAc and VP reward responses intriguing, so we implemented a linear model that fit coefficients for the outcomes on the previous six trials in a manner similar to the Bayer and Glimcher reference the reviewer provided. We did not find a significant impact of trials beyond the most recent in each region, but we added the results of this analysis to Fig 4 because we believe that they show rather succinctly and quantitatively the impact of previous trials in each region (we also provide it here). Perhaps a task that varied the volume of reward (as used in Bayer 2005) would be a more robust approach to evaluate the impact of previous outcomes than our task, which has only two outcomes, but in this task, at least, we feel confident that the strength of the reward signal in NAc and VP is only impacted by the current and previous trials. We thank the reviewer for the illuminating suggestion.

Figure 4. Previous reward outcome impacts current reward firing. (a,c) Normalized activity of all neurons in NAc (a) and VP (c) on sucrose (orange) and maltodextrin (pink) trials of each reward, separated by reward outcome on preceding trial; darker lines indicate that sucrose was the prior trial's reward. Asterisk indicates a significant main effect of previous reward on normalizing firing rate in VP ($F(1,1724) = 10.1$, $p = 0.022$) for the epoch bounded by the vertical dashed lines. (b,d) Normalized reward-related activity of every individual neuron in NAc (b) and VP (d) on trials with each combination of previous and current reward. (e) Mean coefficient weights for the impact of the current and previous 6 trials on normalized firing rate in the same epoch as (a,c) for each neuron in NAc (purple), VP (green), and corresponding data for each neuron with the outcomes shuffled (black) for each region. Error bars are SEM. Asterisks are $p < 0.05$ for Tukey tests comparing VP coefficients to shuffled data, corrected for multiple comparisons. (f) Proportion of the neural populations in VP (green), NAc (purple), and corresponding shuffled neurons (black) with significant coefficients for each of the relative trials. Asterisks represent $p < 0.05$ for chi-square tests on both the distribution of neurons across all four conditions (true and shuffled data from each region) and across the true data from each region.

B. The analysis described in A brings to mind the question of whether a recent history effect would be stronger in NAc neurons if more past trials were included. For instance, suppose the animal obtains 4 maltodextrin rewards in a row followed by a sucrose reward. Would the reward response of NAc neurons to the sucrose reward be different than on sucrose trials preceded by 4 sucrose rewards? Looking just one trial back might be insufficient to pull out such an effect.

Response: This is also a good point but, unfortunately, our sessions were not long enough to permit rigorous analysis of such long chains of reward outcomes. At best we had 30 trials of each reward, 15 of each reward and previous reward combination, 7 for three outcomes, 3 for four outcomes, and only 1 or 2 trials (if any) of four maltodextrin followed by sucrose as the reviewer suggests. We believe that our implementation of the linear model that the reviewer suggested above answers this question to the best of our ability with the current dataset.

C. Once the authors have established the recency parameters that influence selectivity of reward delivery-evoked firing, the next question is whether there is a recency effect in any other form of neural activity observed in this task – for instance, firing evoked by cue presentation and port entry.

Response: As the reviewer suggested, we performed the same analysis for reward responses to that of cue- and port entry-evoked responses, which is summarized in new Supplementary Fig. 8, which we have included here. We found no impact of previous outcomes on cue firing, but we did find an impact of the prior two trials on the VP port entry response (particularly in reward-selective neurons), an unexpected finding that we now mention in the results section and discussion as this signal may relate either to an expectation of reward outcome or to a report of recent reward history. We thank the reviewer for the suggestion to look at the impact of previous outcomes on these events.

Supplementary Figure 8. Impact of previous reward outcomes on cue- and port entry-evoked firing. (a,b) Normalized cue-evoked activity of all neurons in NAc (a) and VP (b) on trials with sucrose (orange) or maltodextrin (pink) as the most recent reward outcome. Blue lines indicate epoch selected for analysis of the impact of previous outcome in (c,d) (0-0.5s from cue onset). (c) Mean coefficient weights for the impact of the previous 6 trials on normalized firing rate for each neuron in NAc (purple), VP (green), and corresponding data for each neuron with the outcomes shuffled (black) for each region. Error bars are SEM. (d) Proportion of the neural

populations in VP (green), NAc (purple), and corresponding shuffled neurons (black) with significant coefficients for each of the relative trials. **(e,f)** As in (a,b) for port entry-evoked activity. Blue lines indicate epoch selected for analysis of the impact of previous outcome in (g,h) (-0.6 - 0.7s from port entry). Asterisk indicates significant effect of previous reward on port entry-evoked activity in VP ($F(1,862) = 16.7$, $p = 0.00048$) **(g)** As in (c), for port entry activity. Asterisks are $p < 0.05$ for Tukey tests comparing VP coefficients to shuffled data, corrected for multiple comparisons. **(h)** As in (d), for port entry models.

D. The reward context-dependent firing could play a role in positive and negative contrast behavioral experiments. It would be useful for the authors to discuss such experiments and whether a potential role for the VP in contrast effects has been explored.

Response: To our knowledge, the role of VP in contrast effect has not been explored, and this is a compelling future direction for researching a role for this signal in a behavioral response. We have added to the discussion the possibility that VP could be involved in the contrast effect.

E. Recency effects are not described for the data in fig. 6 (delivery of 3 different rewards) even though the data set appears to be rich enough to support such an analysis.

Response: This would be an intriguing place to look for the impact of previous outcome on VP firing, but, unfortunately, due to the pseudorandomness of trial presentation, we often have only three trials of a certain current-previous reward combination. We have included what we have here, but we do not feel confident enough in the results due to low trial numbers to include them in the paper. Notably, we also implemented a linear model approach with these data, and we saw no impact of previous outcome, which we attribute mostly to a lack of power to look at this effect rather than a true absence of an impact of reward history.

Response Letter Figure 2. Impact of previous reward outcomes on reward-evoked firing in VP during sessions with three outcomes. Top: Mean firing of reward-selective neurons in VP in response to sucrose, maltodextrin, and water according to the previous outcome. Bottom: Mean coefficients (left) and number of neurons with significant coefficients (right) for linear models fitting the impact of the current and each of the preceding 6 trials on the reward-evoked firing on the current trial for all VP neurons from sessions with three outcomes ($n = 254$).

4. *The authors should discuss the caveat that VP and NAc recordings were made in different animals and therefore in different sessions, and describe what might be gained from making the recordings in the same animals.*

Response: Conducting recordings in different animals rather than in the same animals is an important caveat, and we have added this point to the discussion. Thank you for the suggestion.

5. *I suggest the authors change the title of the paper to reflect the differences between NAc and VP.*

Response: We appreciate this point and have changed the title to summarize the important differences we found in the two regions.

Reviewer 3:

We are happy that Reviewer 3 noted that our “intriguing” results “will make a **substantial contribution to our understanding of NAc and VP reward encoding,**” particularly in helping to “resolve inconsistent reports about the serial vs parallel reward processing in NAc and VP.” We also appreciate that the reviewer found our approach to decoding reward identity from the neural signals in each region “commendable.”

First, to address the reviewer's comments on the rigor of our analysis, we revisited and reorganized all of our analysis and code (this is also in preparation for open source posting in alignment with *Nature Communications* recommendations). This included reviewing spike sorting, resulting in a slightly reduced total number of neurons recorded in each region due to more stringent qualifications for a well-isolated unit. Additionally, we implemented a more conservative method of identifying neurons as reward-selective in order to reduce noise as much as possible, leading to a smaller proportion in each region qualifying as reward-selective. Critically, our detailed and overall pattern of results is the same. We address the specific concerns that the reviewer detailed below.

Major points:

The first major point is that the degrees of freedom of the error in ANOVA analyses are very high (number of observations in analysis: ~3300 for lick data (from 63 sessions), ~2600 for neural activity, and ~81,000 for decoder accuracy). For both lick and neural data analyses, can the authors clarify whether they are using bin, trial, or session as their n? Using multiple bins (per trial across many trials per session) as the number of observations in the analysis would be problematic in that it inappropriately inflates the number of observations and statistical power. It is also unclear whether repeated measures are being accounted for in the analyses presented.

Response: Generally, the reason that we had such high degrees of freedom is that we performed stats on raw data rather than summarized data as much as possible to factor in the variability across trials, sessions, and rats (rather than just finding the average value for each session and performing stats on this number) and then included the random effects of session and subject to account for non-independence in the behavioral and neural data. We have clarified in the methods what the n is for each test we performed. For behavior, n is trial. For example, 63 sessions * ~60 trials per session brings us to the 3300 df the reviewer notes for lick data (we also clarified this difference by representing the lick traces of individual rats in new Supplementary Fig. 2). For the other tests, which compare the activity across different groups of neurons and generally use each neuron's data as the n, we revisited our statistical approaches and, in many cases, we revised our tests to include the random effect of subject, which now accounts for non-independence in the neural data due to subject variability rather than variability across shuffled vs true data or across regions. We have made an effort to more carefully explain each test in the methods. For all pairwise comparison we perform, we correct for multiple comparisons using Tukey tests, which we have made more explicit in the methods. We thank the reviewer for their interest in the rigor of our statistics, and we are open to adjustments in our approach should the reviewer have additional suggestions.

A related point applies for z-score data (lines 647-652). For the z score calculation each bin of interest (variable "optimal" ~20-100 ms) is compared to a 10 s pre-cue baseline period. Please clarify: are the Fmean and Fsd for the baseline period calculated where n is trial (ie many 20-100 ms bins averaged across 10 s to have a per trial value) or are these calculated where n is bin, thus many hundreds of bins inflate number of observations going into the baseline mean/sd calculations? Please report if there is a correction for multiple comparisons.

Response: The n in this case is trial. F_{mean} and F_{sd} are calculated independent of any bins; they are simply a count of the number of spikes in that 10s baseline window for each trial. We did not perform any statistics on the normalized PSTHs so multiple comparisons were not required. The PSTHs were only used for visualization, a point of clarification we have added to the text.

The second major point is that the decoder approach does not take into account different proportions of reward responsive neurons in NAc and VP, and thus the conclusions drawn from this powerful strategy, may not reflect real regional differences in reward responses in NAc and VP. The authors report that a significantly larger proportion of VP neurons (63%) are reward responsive compared to NAc (40%). Further, in peak analyses (for which much of the decoder approach focuses) the authors report 37% VP vs 14% NAc. Thus, a strategy that uses random selection of neurons across the entire population is more likely to randomly select a reward responsive VP neuron than a reward responsive NAc neuron. Therefore, the VP population may appear to be better able to decode reward identity. The authors could instead use only the reward-related neurons to do the decoding, or if this strategy doesn't work due to different numbers of neurons recorded, the authors could employ a normalization strategy, such that the likelihood of randomly selecting a reward responsive neuron is similar for VP and NAc. Alternatively, (though less ideally) authors could do a post-hoc analysis of the presented decoder analyses showing no difference in number of reward selective neurons randomly represented in VP and NAc simulations, to build a case that the data presented is evidence of superior VP decoder accuracy.

Response: We appreciate this point about which neurons are included in ensemble decoding analysis and how this could affect the interpretation. Part of our rationale for performing the decoding analysis was that we wanted an approach that would analyze the amount of reward-specific information in neurons without any initial classification of the neurons as reward-selective. It is likely that randomly selecting ensembles of neurons will lead to a greater proportion of neurons included that are classified as reward selective in Fig. 2, but we still believe this is a useful way of pointing out that VP as a whole has more reward selective information. A separate question is whether reward-selective neurons in VP contain more reward-specific information than those in NAc, which we have addressed with the reviewer's first suggestion of performing the decoding only on reward-selective neurons. This analysis revealed that even among these neurons, VP models are more accurate than those in NAc, suggesting that not only are there more reward-selective neurons in VP, they also discriminate between the two rewards better. We believe this is a very interesting point and we have added it to the paper in a supplementary figure (Supplementary Fig. 7) and also provided it here. We thank the reviewer for the suggestion to pursue this question.

To confirm the reliability of our pseudoensemble decoding data, we increased the number of repetitions of the analysis at each level from 20 in the first manuscript to 50 (so we take 50 random selections of 10, 25, 50, 100, 150 neurons), lending even more confidence in the reliability of this result.

Supplementary Figure 7. Decoding trial identity with only reward-selective neurons.

(a) Average cross-validated decoding accuracy relative to reward delivery time, determined using linear discriminant analysis models trained on spiking data of individual reward-selective neurons (as identified in Fig. 2) across 600ms overlapping bins. Decoding accuracy for NAc (purple), VP (green), and data with shuffled trial identity from each region (black). Shading is SEM. Purple (NAc) and green (VP) lines indicate consecutive bins where accuracy exceeds 99% confidence interval of corresponding shuffled data. (b) Cumulative distribution of accuracies in the bin with the greatest average accuracy in each region (centered at 1.6s in NAc and 1s in VP) and the corresponding shuffled data from that bin in each region. (c) Average cross-validated decoding accuracy relative to reward delivery time of linear discriminant analysis models trained on spiking data of 20 randomly selected groups of 10 or 25 neurons in NAc across 600ms overlapping bins and corresponding models trained on data with trial identity shuffled. Shading is SEM. (d) Same as (c) for VP pseudoensemble models of 10, 25, 50, 100, or 150 neurons. (e) Average accuracy of each replicate for the bin with peak accuracy for each pseudoensemble size in each region. Asterisk indicates significant main effect of region on accuracy for 10 and 25 neuron ensembles ($F(1,196) = 38.9$, $p = 2.7E-9$). (f) Average peak accuracy time post-reward for each replicate of each pseudoensemble size in each region. Asterisk indicates significant main effect of region on time of peak accuracy for 10 and 25 neuron ensembles ($F(1,196) = 54.8$, $p = 3.8E-12$).

Minor points:

Could the authors please clarify whether the stats done on raw or smoothed data?

Response: All stats were performed on unsmoothed data. We have added this point to the methods.

Fig 3: Shuffled VP or NAC and why get to ~70% accuracy? For peak analyses, how did authors determine peak bin for shuffled data? Is it 1.0 s for VP and 1.7 s for NAc?

Response: The reviewer is correct. We used the corresponding shuffled bin for each of the true data bins with peak accuracy, which we deemed the most appropriate comparison. We have clarified this in the text. Accuracy occasionally gets to 70% because it is computed from the average of 5 sets of cross-validated classification of left-out data. Although rare, it is possible to occasionally classify the data correctly by chance, which is why the accuracy for a shuffled model for an individual unit may get to 70%. This is why we consider it important to compare the true data to shuffled data to be as rigorous as possible.

Fig 3 Is Shuffled vs true a subtraction score? Bin minus relative bin?

Response: Shuffled vs true is the comparison of the shuffled data to the true data within the ANOVA because both the shuffled data and the true data are included in the test. Like the effect of “reward” compares sucrose and maltodextrin data, the effect of “shuffled vs true” compares the shuffled and true data. We have clarified this in the text.

Fig 4 stats ANOVA on Z-score or raw data? What is sig. z-score value?

Response: We performed this ANOVA on Z-score data. We did this to normalize the values to comparable levels since we are looking at an entire population of neurons with a variety of firing rates. There was no significance value for the Z-score; all neurons were included, and the test for significance came from the ANOVA.

Fig 5 line 244 (text) comparison is unfair to make due to only a subset of rats being used. A better comparison would be to compare proportion of reward responsive in these two rats across two dif types of reward sessions.

Response: We appreciate this point and we changed the comparison to only look at the sucrose vs maltodextrin data from these two rats. Thank you for pointing this out.

Could not find stats for 5D.

Response: The neurons were pre-selected because they met the statistical criteria for reward selectivity. Rather than report the results of statistical tests for every individual neuron, we conducted an additional statistical test on grouped data and found a significant interaction between reward and number of trials, which we now describe in the methods and the results.

Could not find statistical analyses for figure 6 data

Response: The figures are a summary of neurons that passed the statistical criteria for reward selectivity. Nevertheless, to be thorough, we ran an ANOVA on the firing rates of the sucrose-

preferring neurons (in 6c) and found significantly different firing for all three rewards (corrected for multiple comparisons), which we have now added to the methods and the results.

Y-axis in Fig. 5A is not labeled.

Response: We added a Y-axis label, thank you.

Sup fig 5 is an excellent resource.

Response: Thank you.

REVIEWERS' COMMENTS:

Reviewer #1 (Remarks to the Author):

The revision addressed most of my concerns.

As the new title indicates, the main conclusion of this article is that VP encodes relative reward value earlier than NAc.

In this respect, the authors mentioned the importance of common inputs to the VP and NAc. Another possible interpretation is that cortical inputs originating from the medial prefrontal cortex enter the ventral pallidum through the cortico-subthalamo-pallidal pathway in addition to the cortico-striato-pallidal pathway. Based on conduction velocities from the existing literature, cortical stimulation can evoke the disynaptic excitation in VP earlier than the monosynaptic excitation in NAc.

I suggest the authors to include this possibility in the section of Discussion.

Reviewer #2 (Remarks to the Author):

The authors have satisfied my concerns. I just have two minor corrections:

- 1. The legend for fig. S7c shows 6 data groups, but only 3 are shown.**
- 2. Line 209: delete the word "some".**

Reviewer #3 (Remarks to the Author):

The authors have been very thorough in addressing my concerns by reanalyzing and adding new neural analyses that further strengthen their interpretations of VP and NAc relative reward encoding. The added detail and clarifications on how the data were analyzed are much appreciated. I have no further comments or concerns.

Response to reviewers

Reviewer 1:

The revision addressed most of my concerns. As the new title indicates, the main conclusion of this article is that VP encodes relative reward value earlier than NAc. In this respect, the authors mentioned the importance of common inputs to the VP and NAc. Another possible interpretation is that cortical inputs originating from the medial prefrontal cortex enter the ventral pallidum through the cortico-subthalamo-pallidal pathway in addition to the cortico-striato-pallidal pathway. Based on conduction velocities from the existing literature, cortical stimulation can evoke the disynaptic excitation in VP earlier than the monosynaptic excitation in NAc. I suggest the authors to include this possibility in the section of Discussion.

Response: We are glad that our revisions satisfied most of the reviewer's concerns. We have revised the following sentences in the discussion to address this possibility of the cortico-subthalamo-pallidal pathway producing the earlier value encoding in VP.

Together, our findings support the notion that VP processes certain aspects of reward independently of NAc, and they highlight the importance of studying other inputs to VP that could provide the input for the rapid, phasic reward-specific signal observed in VP here. Candidate regions include amygdala^{51,52}, lateral hypothalamus^{53,54} and prefrontal cortex, which, in addition to direct projections⁵⁵, could provide input via the subthalamic nucleus^{56,57}, a route that is reported to be faster than through striatum^{58,59}.

Reviewer 2:

The authors have satisfied my concerns. I just have two minor corrections:

- 1. The legend for fig. S7c shows 6 data groups, but only 3 are shown.*
- 2. Line 209: delete the word "some".*

Response: We are glad that our revisions satisfied the reviewer's concerns. We have made these two corrections.

Reviewer 3:

The authors have been very thorough in addressing my concerns by reanalyzing and adding new neural analyses that further strengthen their interpretations of VP and NAc relative reward encoding. The added detail and clarifications on how the data were analyzed are much appreciated. I have no further comments or concerns.

Response: We are very glad and grateful that the reviewers' suggestions led to stronger interpretation. Thank you to all.